# A novel source of arterial valve cells linked to bicuspid aortic valve without raphe in mice

Lorriane Eley[1], Ahlam MS Alqahtani[1], Donal MacGrogan[2,3], Rachel V Richardson[1], Lindsay Murphy[1], Alejandro Salguero-Jimenez[2,3], Marcos Sintes Rodriguez San Pedro[1], Shindi Tiurma[1], Lauren McCutcheon[1], Adam Gilmore[1], José Luis de La Pompa[2,3], Bill Chaudhry[1], Deborah J Henderson[1]*

[1]Institute of Genetic Medicine, Cardiovascular Research Centre, Newcastle University, Newcastle upon Tyne, United Kingdom; [2]Intercellular Signalling in Cardiovascular Development and Disease Laboratory, Centro Nacional de Investigaciones Cardiovasculares Carlos III, Madrid, Spain; [3]Centro de Investigación Biomédica en Red de Enfermedades Cardiovasculares, Instituto de Salud Carlos III, Madrid, Spain

**Abstract** Abnormalities of the arterial valve leaflets, predominantly bicuspid aortic valve, are the commonest congenital malformations. Although many studies have investigated the development of the arterial valves, it has been assumed that, as with the atrioventricular valves, endocardial to mesenchymal transition (EndMT) is the predominant mechanism. We show that arterial is distinctly different from atrioventricular valve formation. Whilst the four septal valve leaflets are dominated by NCC and EndMT-derived cells, the intercalated leaflets differentiate directly from *Tnnt2-Cre+/ Isl1+* progenitors in the outflow wall, via a Notch-Jag dependent mechanism. Further, when this novel group of progenitors are disrupted, development of the intercalated leaflets is disrupted, resulting in leaflet dysplasia and bicuspid valves without raphe, most commonly affecting the aortic valve. This study thus overturns the dogma that heart valves are formed principally by EndMT, identifies a new source of valve interstitial cells, and provides a novel mechanism for causation of bicuspid aortic valves without raphe.

DOI: https://doi.org/10.7554/eLife.34110.001

*For correspondence:
deborah.henderson@ncl.ac.uk

Competing interests: The authors declare that no competing interests exist.

## Introduction

Bicuspid aortic valve (BAV) is the commonest congenital cardiac malformation in man (*Hoffman and Kaplan, 2002*). Together with stenosis of the aortic and pulmonary valves, BAV commonly accompanies other congenital cardiovascular malformations, for example, ventricular septal defects, double outlet right ventricle, hypoplastic left heart syndrome and aortic coarctation (*Siu and Silversides, 2010*). BAV is also associated with late onset aortopathy (*Tadros et al., 2009*). These associations may reflect common developmental origins and/or abnormal haemodynamic forces, but little is known specifically about arterial (semilunar) valve development and mechanisms have been extrapolated from tricuspid and mitral (atrioventricular) valve development (*de Vlaming et al., 2012*). Similarly, the pathogenesis of stenotic, dysplastic or bicuspid arterial valves remains obscure, although fusion of adjacent leaflets in fetal life, resulting in the presence of a raphe, is presumed to cause many cases of BAV.

Over the past 20 years we have developed a good understanding of how the outflow tract develops. Cells derived from the second heart field (SHF) make the major contribution to the outflow tract walls, forming myocardium in the proximal ventricular part, smooth muscle cells in the distal arterial

walls, and also the inner endocardial lining (*Kelly et al., 2014*). In the early unseptated outflow tract, peristaltic contraction propels blood forward and retrograde flow is prevented by occlusion of the heart tube lumen by voluminous cardiac jelly interposed between endothelium and contracting myocardium (*Nomura-Kitabayashi et al., 2009*). As the outflow tract lengthens, the cardiac jelly becomes localised to discrete areas, endocardial cushions, which are the anlagen of the mature valve leaflets (*de Vlaming et al., 2012*; *Phillips et al., 2013*). These initially acellular endocardial cushions become populated with mesenchymal cells that give rise to the valve interstitial cells. A major source of cells in both atrioventricular and arterial valves is epithelial or endothelial to mesenchymal transformation (EMT or EndMT; [*Markwald et al., 1977*; *Markwald et al., 1978*; *de Vlaming et al., 2012*]). In addition, the main outflow tract cushions, that fuse to separate aortic and pulmonary blood flow and also give rise to the arterial valve leaflets, receive a major contribution of neural crest cells (NCC) that have migrated through the pharyngeal arches (*Kirby et al., 1983*; *Waldo et al., 1999*; *Jiang et al., 2000*; *Jain et al., 2011*; *Phillips et al., 2013*). In contrast few NCC are seen in the atrioventricular valves. Smaller swellings located between the main superior and inferior outflow tract cushions, commonly called intercalated cushions (IC), give rise to the remaining two leaflets (*Kramer, 1942*; *Anderson et al., 2012*; *Sizarov et al., 2012*; *Yang et al., 2013*). We have previously shown that NCC are essential for positioning the main outflow tract endocardial cushions and patterning the arterial valve leaflets, but make only a minor contribution to these IC (*Phillips et al., 2013*). More recently, we have also shown that the arterial roots (*Figure 1A*) that connect the ventricles with the great arteries and house the arterial valves, have a complex origin from both SHF-derived cells and NCC (*Richardson et al., 2018*). However, the contribution of each lineage to the different valve leaflet primordia, and how these relationships affect normal and abnormal leaflet patterning including BAV, remains obscure.

This study demonstrates how the arterial valve leaflets form and defines the origins of the cells that populate these leaflets. We show that despite the mature arterial valve leaflets appearing indistinguishable, the IC have an entirely distinct origin and underpinning developmental mechanisms from the major cushions, originating from a novel, direct SHF contribution that is independent of endMT.

## Results

### The intercalated cushions are morphologically distinct from the main outflow cushions

Although the presence of the IC and their participation in arterial valve leaflet development is well-recognised (*Kramer, 1942*) (*Figure 1B*), their origin and composition have not been established. We therefore analysed their development from mouse embryonic day (E)9.5 and human Carnegie stage (CS)14. At E9.5, the outflow of the heart consists of a simple tube with an outer myocardial wall, an inner layer of endocardium and an intervening acellular space filled with cardiac jelly. There was no evidence of discrete main or intercalated cushions (*Figure 1C*). By E10.5 the IC were first seen as anterior and posterior thickenings of the outflow tract outer wall, bulging into localised expansions of cardiac jelly heralding the main outflow tract cushions (*Figure 1D* and *Figure 1—figure supplement 1*). The IC were obvious at E11.5 as compact swellings within the distal outflow tract wall (*Figure 1E*) whilst rotation of the outflow tract (*Bajolle et al., 2006*) placed the aortic IC to the right and the pulmonary IC to the left. By E12.5, all six primordia of the forming aortic and pulmonary valve leaflets appeared morphologically similar, although packing of cells in the forming IC appeared more dense (*Figure 1F*). At E13.5 and E15.5 the leaflets were indistinguishable from each other using standard histological staining (*Figure 1G,H*). These observations were confirmed in human embryos at equivalent developmental stages (*Figure 1I–L*).

Although of similar appearance, the pulmonary IC became evident earlier in development and was more discrete than the aortic IC at E10.5-E11.5. Moreover, the two IC were positioned at different levels within the outflow reflecting the orientations of the adult valve structures. Thus, the pulmonary IC sits more distally by E11.5 and is surrounded laterally by a cuff of outflow tract myocardium (*Figure 1M,P,R*), whereas the larger aortic IC only has proximal contact with myocardium (*Figure 1N,Q,R*). Importantly, both IC have distal continuity with undifferentiated (non-myocardial) outflow tract wall (*Figure 1O,Q* and *Figure 1—figure supplement 1*).

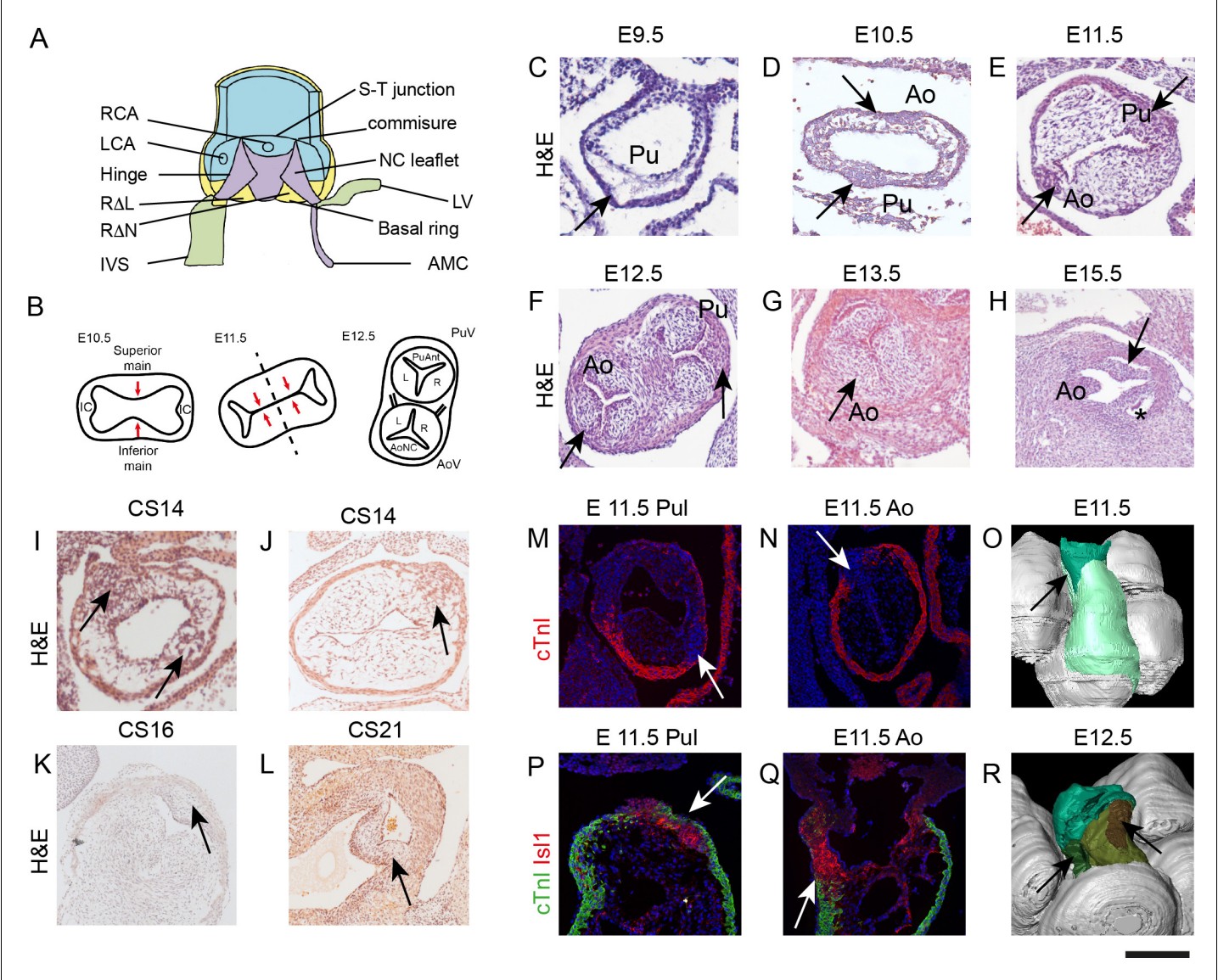

**Figure 1.** IC have a different morphology to the main outflow cushions during development. All sections, except P,Q are taken in a frontal orientation. Arrows point to the IC or the IC-derived leaflet primordia, depending on stage. Images are typical examples from a minimum of n = 3 at each stage. (**A**) Diagram of the aortic root showing the semilunar attachment pattern of the arterial valve leaflets. The colours reflect the tissue composition of the root (green = myocardium, blue = smooth muscle cells, yellow = mural fibrous tissue, purple = valvular fibrous tissue). (**B**) Cartoon (based on [**Kramer, 1942**]) illustrating how fusion of the main outflow cushions together with growth of the IC produces the three arterial valve primordia. (**C–H**) At E9.5 (**C**), the outflow tract is a simple tube with an outer myocardium and inner endocardium, but with few cells in between. The arrow points to the region where the IC will appear, but there is no evidence of wall thickening at this stage. By E10.5 (**D**), cells are seen in the cardiac jelly and IC appear as bulges in the outflow wall (arrows). At E11.5 (**E**), the IC (arrows) have expanded but are significantly smaller than the main outflow cushions. The posterior valve primordia (arrows) are well developed and appear similar to the right and left valve primordia at E12.5-E13.5 (**F,G**), although the cells appear more densely packed. By E15.5 (**H**), the valve sinuses have formed and the leaflets are apparent. (**I–L**) IC are also apparent in stage matched (CS14-16) human embryos and have a similar morphology to those seen in mouse at CS21. (**M–Q**) The pulmonary IC (labelled red by Isl1 in P) is encased by a thin layer of cTnI-expressing (green) myocardium (arrows in M,P), whereas the aortic IC is not (**N,Q**). At E11.5, the IC are in continuity with Isl1-expressing cells in the pharyngeal region (red in P,Q). (**O,R**) At E11.5 (**O**), 3D reconstruction (modified from those first presented in [**Anderson et al., 2012**]) shows the non-myocardial component of the outflow (dark green) extends into the myocardial component (light green). The IC form within these extensions (arrows). At E12.5, the pulmonary IC (brown, see arrow) is found more distally than the aortic IC (green, see arrow). AMC – aortic mitral continuity; Ao – aorta; AV – aortic valve; IC – intercalated cushion; IVS – interventricular triangle; LCA – left coronary artery; LV – left ventricular myocardium; NC – non coronary; P or Pul – pulmonary trunk; PV – pulmonary valve; RCA – right coronary artery; RΔL – right-left interleaflet triangle; RΔN – right - non-coronary interleaflet triangle; S-T – sino-tubular. Scale bar: C-E,I-K,P,Q = 100 μm, F,G = 150 μm, H,L = 200 μm.

DOI: https://doi.org/10.7554/eLife.34110.002

*Figure 1 continued on next page*

*Figure 1 continued*

The following figure supplement is available for figure 1:

**Figure supplement 1.** Histological appearance of the IC as they first form.

DOI: https://doi.org/10.7554/eLife.34110.003

## IC undergo late differentiation to become valve primordia

These initial observations suggested that the origins of the IC are developmentally distinct from the main outflow cushions. To evaluate the tissue characteristics of these primordia we examined the expression of chondroitin sulphate, a component of both the extracellular matrix (ECM) of the cardiac jelly and valve primordia (*Markwald et al., 1978*), using CS56 antibody. We also investigated expression of cardiac troponin I (cTnI), a myocardial specific protein, to elucidate the relationships of the IC with the myocardial wall. At E10.5 cardiac jelly, labelled by CS56 antibody, was found in a circumferential layer between the myocardium and endocardium, with laterally located accumulations of cells that prefigured the main cushions (*Figure 2A*). Although the IC within the wall were not labelled by CS56, a thin layer of CS56-expressing cardiac jelly overlaid their luminal surface (*Figure 2D,G*). The outflow tract wall labelled with cTnI antibody, but despite continuity with this wall, the IC did not express cTnI. At E11.5, the main outflow tract cushions contained many cells and continued to strongly label with CS56 antibody (*Figure 2B*). In contrast, although the IC had expanded, they remained devoid of labelling for cTnI or for CS56, except for the thin layer of adluminal cardiac jelly (arrowheads in *Figure 2E*). When examined at E12.5, the main cushions and the IC labelled with the CS56 antibody (*Figure 2C,F,I*). To further understand the differentiation of these IC we examined them for the expression of proteins relevant to endocardial cushion differentiation. Versican is a chondroitin sulphate proteoglycan that is expressed early and plays essential roles in cushion formation (*Henderson and Copp, 1998*; *Mjaatvedt et al., 1998*). At E10.5, Versican was only detected in the cardiac jelly of the main septal cushions and the thin layer of cardiac jelly overlying the IC. However, at E11.5, prior to expression of CS56, and also at E12.5, Versican was expressed within the bulk of the IC (*Figure 2J–O*). At E10.5 Sox9, a transcription factor necessary for the formation of endocardial cushions and maintained in mature valve leaflets (*Akiyama et al., 2004*; *Richardson et al., 2018*), was abundantly expressed in the circumferential layer of cardiac jelly where the main cushions form, but only in the area immediately adjacent to the endothelium overlying the IC that was labelled by CS56 (*Figure 2P,S* – compare to 2 D,G); there was no Sox9 in the main part of the IC at this stage. At E11.5 Sox9 was also expressed throughout the IC and this was maintained at E12.5 (*Figure 2Q,R,T,U*). Taken together, these findings indicate that at early stages the IC are overlaid by a layer of cardiac jelly but are not endocardial cushions in the classical sense. However, as they mature they express markers of valve interstitial cells. In view of this we henceforth refer to them in this manuscript as intercalated valve swellings (ICVS) as in Kramer's original description (*Kramer, 1942*).

## Undifferentiated SHF cells directly contribute to the ICVS

We have previously shown that whilst NCC and EndMT-derived cells (EDC) make a major contribution to valve primordia derived from the main outflow cushions at E11.5 and E12.5, they make only a minor contribution to the laterally-placed IC (*Phillips et al., 2013*). These patterns are highly reproducible (n > 10). More thorough analysis showed that the proportion of *Wnt1-Cre+* lineage cells was significantly different between the valve primordia, $X^2$ (df5, N = 6842)=959.4, p<0.0001 (*Figure 3— source data 1*). Pairwise chi-squared comparison, with Bonferroni correction for multiple testing, confirmed that the aortic posterior (non-coronary) and anterior pulmonary primordium were statistically different from all other primordia (for each, p<0.0002). Whereas the left aortic primordium was statistically different from the remaining primordia (p<0.02), the aortic right, pulmonary left and pulmonary right were not significantly different with respect to the number of *Wnt1-Cre+* lineage cells they contained. Chi squared analysis showed that the proportion of *Tek-Cre* lineage cells between the different arterial valve primordia were significantly different, $X^2$ (df5, N = 7389)=782.3, p<0.0001 (*Figure 3—source data 1*). Pairwise Chi-squared comparison between the primordia, with Bonferroni correction for multiple testing, confirmed that the aortic non-coronary, the pulmonary right and the pulmonary anterior were not statistically different in the proportion of *Tek-Cre+* lineage cells

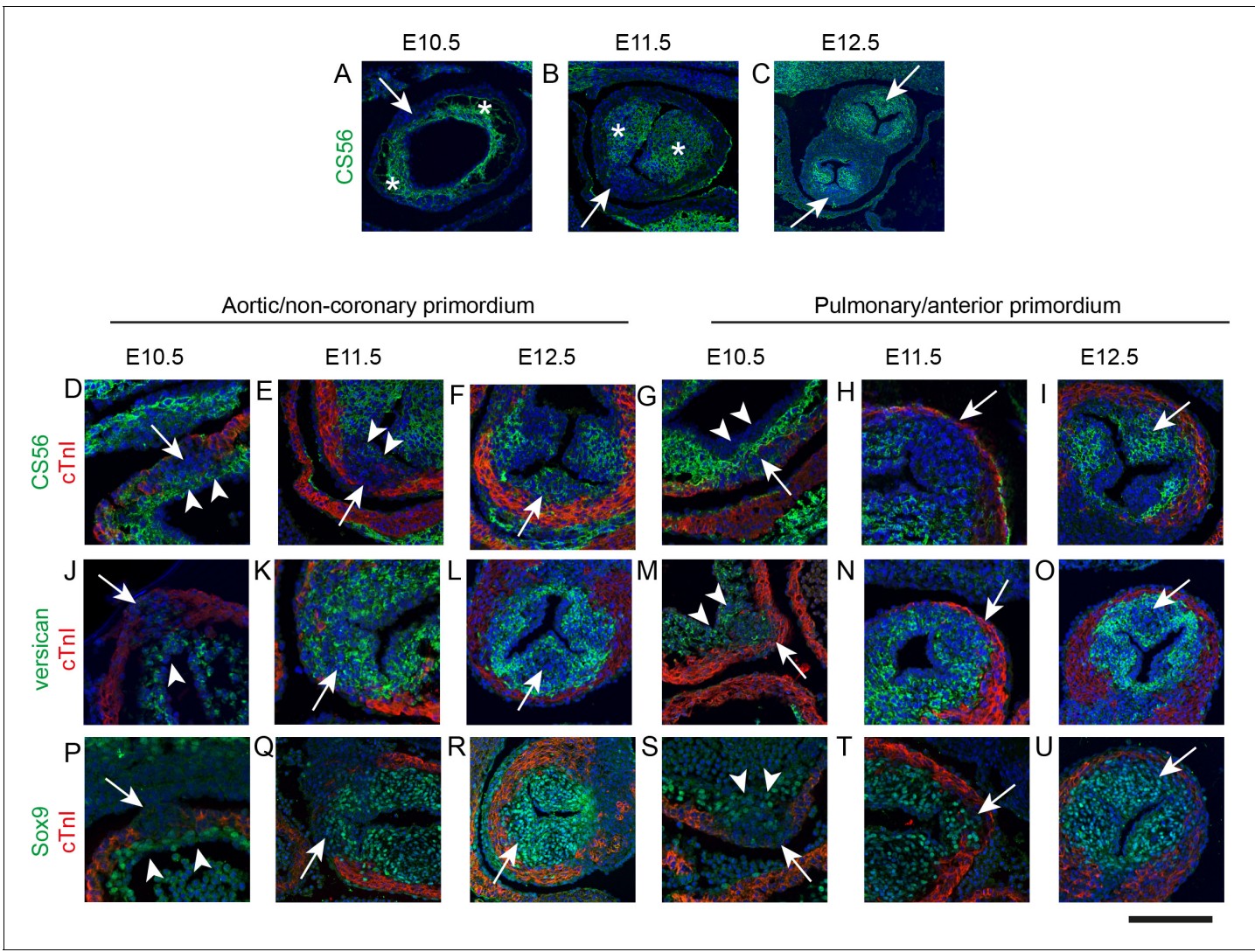

**Figure 2.** IC are molecularly distinct from the main outflow cushions. In each case, arrows mark the IC. * mark the main outflow cushions. In D-U the myocardium is labelled with cTnI (green). Images are typical examples from a minimum of n = 3 at each stage. (A–C) The cardiac jelly of the main outflow cushions contain chondroitin sulphate molecules, recognised by the antibody CS56, from E10.5 to E12.5, whereas these are only abundant in the IC at E12.5. (D–I) Higher power of CS56 immunoreactivity shows that although the bulk of the IC stain with neither cTnI nor CS56 at E10.5-E11.5 (arrows), they are overlaid by a layer of CS56-expressing material (arrowheads in D,E,G) that is continuity with the main outflow cushions. By E12.5, the majority of the leaflet primordium is labelled by CS56. (J–O) Versican is expressed in the cardiac jelly of the main cushions at E10.5 and the adluminal region overlying the IC (arrowhead in J,M). Versican is not found in the bulk of the IC at E10.5, (J,M) but is expressed more broadly in the IC at E11.5 and E12.5. Notably, versican is missing from the centre part of both the aortic and pulmonary IC at E12.5 (arrows in L,O). (P–U) Sox9 labels only the layer overlying the IC at E10.5 (arrowheads in P,S) but is expressed throughout the ICVS by E11.5 and at E12.5. Scale bar: A = 90 µm, B,F,I,L,O,R, U = 120 µm, C = 250 µm, D,G,J,M,P,S = 50 µm, E,H,K,N,Q,T = 70 µm.

DOI: https://doi.org/10.7554/eLife.34110.004

they contained. However, all the remaining primordia were statistically different (p<0.0002). Thus, these data showed that there were marked variations in the NCC and EDC contributing to each leaflet and confirmed that there were insufficient *Wnt1-Cre+* and *Tek-Cre+* lineage traced cells to account for all the cells present in these valve primordia (*Figure 3A–F* and *Figure 3—figure supplement 1*). We therefore looked for alternative sources to explain the origins of the remaining cells in the developing arterial valve leaflets. As the parietal valves of the atrioventricular valves are known to contain epicardially-derived cells (EPDC; [*Gittenberger-de Groot et al., 1998*; *Wessels et al., 2012*]). We asked whether EPDC could contribute to the ICVS or other outflow cushions. To test this, we performed tamoxifen induction in *Wt1-Cre-ERT2* mice starting at E9.5. Analysis at E12.5

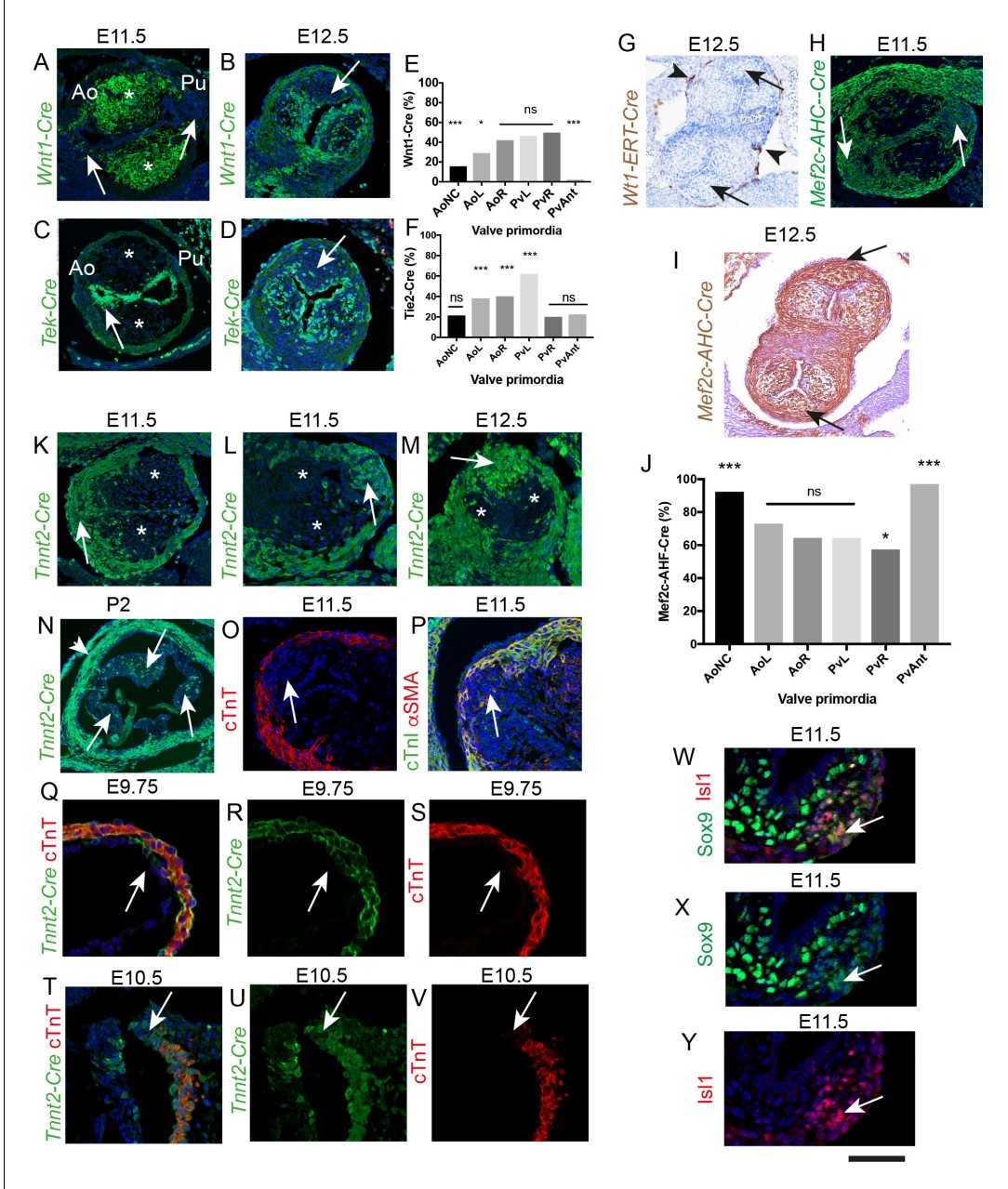

**Figure 3.** The ICVS are formed from undifferentiated SHF-derived cells that are labelled by TnnT2-Cre. All sections are frontal unless otherwise stated. Arrows mark the ICVS. * mark the main outflow cushions. Images are typical examples from a minimum of n = 3 at each stage. (A–F) NCC labelled by *Wnt1-Cre* (A,B) make a significant contribution to the main outflow cushions and right and left valve primordia at E11.5 and E12.5, but make only a minor contribution to the ICVS and anterior leaflets (arrows). *Tek-Cre* labels cells in the endocardium (arrow in C), but few cells in either the main cushions or ICVS at E11.5 (C), although they are abundant in the left and right leaflets by E12.5 (D). Quantification of cells at E12.5 (E,F) confirms these observations and reveals that there are a number of cells in the leaflets, particularly the anterior and posterior derived from the ICVS, that are not labelled by NCC or EDC (data reanalysed from [*Phillips et al., 2013*]). (G–J) *Wt1-ERT-Cre* labels the epicardium (arrowheads in G) but not cells in the ICVS (arrows in G). The ICVS are labelled by *Mef2c-AHF-Cre* at E11.5 and E12.5 (H,I), as are the main cushions and the left and right leaflet primordia. Quantification of these cells (J) shows that the *Mef2c-AHF-Cre*-expressing population is the major contributor to the leaflet primordia. (K–N) Cells expressing *Tnnt2-Cre* label the cells in the ICVS at E11.5 (arrows in K,L) and the posterior valve leaflet primordia at E12.5 (M). A few labelled cells are also seen in the left and right primordia. (N) *Tnnt2-Cre* labels cells in all leaflets (arrows) of the aortic valve at P2, although it is more abundant in the posterior leaflet. *Tnnt2-Cre* also labels the walls of the aortic sinuses that are composed of SMC (arrowhead). (O,P) Antibodies specific for cardiac troponin-T (O), cTnI (green in P) and αSMA (red in P) label the outflow wall but not the ICVS at E11.5. (Q–V) At E9.5, a *Tnnt2-Cre*-positive, but cTnT-negative region (arrows in Q-S) is apparent in the distal outflow wall. A similar region can be seen in transverse sections at E10.5 (arrows in T-V). N.B. *mTmG* was used as the reporter line in Q-S so the GFP staining appears membrane associated in these sections. (W–Y) Some of the cells in the ICVS

*Figure 3 continued on next page*

*Figure 3 continued*

label co-express (yellow; arrows) Isl1 and Sox9 antibody at E11.5. Scale bar: A-D,H,K,L,M = 100 μm, G,I = 150 μm, N = 400 μm, O,p=60 μm, Q-V = 50 μm, W-Y = 40 μm.

DOI: https://doi.org/10.7554/eLife.34110.005

The following source data and figure supplements are available for figure 3:

**Source data 1.** Raw data from lineage tracing for *Wnt1-Cre+*, *Tek2-Cre+* and *Mef2c-AHF-Cre+* cells in E12.5 valve leaflets.

DOI: https://doi.org/10.7554/eLife.34110.010

**Figure supplement 1.** Lineage of cells in the main outflow cushions.

DOI: https://doi.org/10.7554/eLife.34110.006

**Figure supplement 2.** Confirmation of timing of *Cre* expression.

DOI: https://doi.org/10.7554/eLife.34110.007

**Figure supplement 3.** *Tnnt2*-Creexpression in SMC.

DOI: https://doi.org/10.7554/eLife.34110.008

**Figure supplement 4.** Persistence of Isl1 expression in ICVS.

DOI: https://doi.org/10.7554/eLife.34110.009

showed that although the epicardium itself was strongly labelled, there was no contribution of EPDC into the forming arterial valve leaflets (*Figure 3G*).

The dorsal pericardial wall and the pharyngeal mesoderm are composed of SHF cells that directly generate the outflow tract myocardium, much of the smooth muscle and fibrous tissue in the proximal and distal outflow vessel walls, and also the endocardium that lines these structures (*Waldo et al., 2005*; *Harmon and Nakano, 2013*; *Verzi et al., 2005*; *Richardson et al., 2018*). Moreover, the SHF through endothelial EndMT, indirectly populates the outflow tract cushions (*Verzi et al., 2005*); *Figure 3—figure supplement 1*). We wondered whether SHF progenitors might contribute to the ICVS, over and above those provided by EndMT. By lineage tracing using *Mef2c-AHF-Cre*, it was possible to identify all SHF-derived cells adding into the valve anlagen, either directly or through EndMT. Analysis at E11.5 and E12.5 revealed that cells derived from the *Mef2c-AHF-Cre+* SHF make a major and highly reproducible contribution to all of the arterial valve leaflet primordia, including the ICVS (*Figure 3H–J*). Chi squared analysis following quantification of the *Mef2c-AHF-Cre+* lineage cells in the E12.5 leaflet primordia showed that the proportion of these cells were significantly different between the different arterial valve primordia, $X^2$ (df5, N = 7435) =748.2, p<0.0001 (*Figure 3—source data 1*). Pairwise Chi-squared comparison between the primordia, with Bonferroni correction for multiple testing, confirmed that the aortic anterior (non-coronary) primordium was statistically different from all other primordia (p<0.0002), as was the pulmonary posterior primordium (p<0.0002). Moreover, whilst the right pulmonary primordium was statistically different from the remaining primordia (p<0.02); the aortic left, aortic right and pulmonary left were not significantly different. Importantly, our analysis demonstrated that the numbers and distribution of *Mef2c-AHF-Cre+* lineage cells in the ICVS and in both the main cushions were complementary to the *Wnt1-Cre+* lineage (compare *Figure 3J* with 3E) and were sufficient to explain the presence of all cells.

## Origin of ICVS directly from SHF progenitors in the outflow wall

A possible source of the SHF-derived cells within the ICVS was trans-differentiation of outflow tract myocardium. To explore this, we carried out lineage tracing using the cardiomyocyte-specific Troponin T-Cre (cTnT; *Tnnt2-Cre*) line. Analyses of the ICVS, both during formation at E11.5 and when they resemble the main cushions at E12.5, showed that the ICVS and the adjacent outflow tract wall was indeed composed of cells demonstrating *Tnnt2-Cre+* lineage (*Figure 3K–M*). In addition to this major contribution of *Tnnt2-Cre+* lineage cells to the ICVS at these stages, a few labelled cells were also found in the right and left leaflets derived from the main septal cushions, located within the outer parts of the leaflets, close to the myocardial wall. Examination of *Tnnt2-Cre;eYFP* embryos at later stages of development showed that the *Tnnt2-Cre+* lineage cells were found in all three arterial valves throughout gestation and postnatally, although the majority were found in the anterior/posterior leaflets; these data indicate that *Tnnt2-Cre+* lineage cells make a permanent contribution to the valve leaflets (*Figure 3N*). However, as shown earlier, the E11.5 ICVS did not contain differentiated cardiomyocytes, thus did not express cTnT, cTnI, nor αSMA, a smooth muscle cell (SMC) marker

(*Figure 3O,P*). As the *Cre* lineage tracing indicated cells that not only currently expressed cTnT, but also those that were descended from cells that previously expressed *c*TnT, we used antibodies directed against cTnT/cTnI together with *Tnnt2-Cre* lineage tracing at earlier stages of development, to establish whether cells in the ICVS ever expressed these markers. Despite their *Tnnt2-Cre+* lineage labelling, the ICVS did not label with antibodies to cTnT nor cTnI at any stage of development. At E9.75, before the ICVS could be seen in the distal outflow, a region of *Tnnt2-Cre+;cTnT-* labelling was seen, likely indicating where the ICVS would develop (*Figure 3Q–S*). Similarly, At E10.5 the most distal outflow wall was *Tnnt2-Cre*-expressing, but did not label with cTnT antibody (*Figure 3T–V*). Thus, our data argue against trans-differentiation of ICVS cells from cardiomyocytes. To clarify the timing of Cre expression driven by the *Tnnt2* promoter activity in the ICVS, we used an anti-Cre antibody. By examining embryos at E10.5-E12.5 we showed that at E10.5 Cre protein was found in the forming ICVS and in the myocardial wall. However, by E11.5, and continuing at E12.5, Cre protein was absent from the ICVS, although still maintained in the myocardial walls (*Figure 3—figure supplement 2*). Taken together these data indicate that *Tnnt2-Cre* activation in the forming ICVS occurred prior to full differentiation to cardiomyocytes and taking into account perdurance of Cre protein, is likely to occur before E10.5. Interestingly, *Tnnt2-Cre+* lineage cells were also observed in SMC within the arterial wall later in development (*Figure 3N* and (*Figure 3—figure supplement 3*).

A key question is whether these ICVS cells originate directly from undifferentiated SHF cells. As they remain in continuity with the dorsal pericardial wall, we wondered if they might also have retained an undifferentiated state. Islet 1 (Isl1) is a specific marker of multipotent SHF progenitors that are yet to differentiate into a broad range of cell types including cardiomyocytes (*Cai et al., 2003*). At E11.5 the ICVS were clearly delineated within the distal outflow wall by Isl1 antibody (*Figure 3W,Y* and *Figure 3—figure supplement 4*); co-localisation studies showed that some of the cells also expressed Sox9 and thus were directly differentiating into valve interstitial cells (*Figure 3W–Y* and *Figure 3—figure supplement 4*). By E12.5, Isl1 expression had essentially disappeared from the outflow tract, although persisting low level Isl1 expression was noted in a few cells in the ICVS (*Figure 3—figure supplement 4*). Thus, the cells within the ICVS, although of *Tnnt2-Cre*+ lineage, are formed from *Mef2c-AHF-Cre*-expressing SHF progenitor cells that maintain expression of Isl1 for longer than the surrounding cells, suggesting they differentiate later. Co-expression of reducing Isl1 protein levels but increasing Sox9 protein levels, in the setting of absence of protein for differentiated cardiomyocyte markers, supports direct derivation of valve precursors from SHF, rather than trans-differentiation from myocardial wall.

## ICVS derive from SHF epithelium, but not through EMT

We have previously shown how SHF progenitor cells from the dorsal pericardial wall add to the elongating outflow tract at E9.0-E9.5 (*Ramsbottom et al., 2014*) and form the myocardium of the outflow tract. As they do so they initially up-regulate E-cadherin and show marked cellular polarisation. As addition continues, they lose their epithelial phenotype, down-regulate Isl1, and then express markers of differentiated cardiomyocytes in the segment of outflow tract we termed the transition zone (*Ramsbottom et al., 2014*). E-cadherin was expressed throughout the distal outflow wall including the forming ICVS at E10.5, (*Figure 4A–C*) but was not expressed in the main cushions. By E11.5 levels of E-cadherin expression were reduced in the ICVS (*Figure 4D–F*) and by E12.5, E-cadherin was absent (*Figure 4G–I*). This was paralleled by β-catenin expression, which also labels epithelial cells, (*Figure 4J*), but was entirely different from N-cadherin and fibroblast specific protein 1 (Fsp1) immuno labelling, both of which are markers of mesenchymal cells. Specifically, neither N-cadherin nor Fsp1 were expressed in the ICVS at any stage of development (*Figure 4K,L*), indicating that the ICVS do not contain typical mesenchymal cells. Interestingly, collagen I and elastin, both ECM proteins normally seen surrounding SMC in the tunica media of blood vessels and also found in mature valve leaflets (*Hinton and Yutzey, 2011*), were transiently expressed only in the forming ICVS but not in the surrounding walls or main cushions at E11.5 (*Figure 4M–O* and *Figure 4—figure supplement 1*). These data suggest that at E11.5 and E12.5, the ICVS cells express different markers to the cells that make up the right and left leaflets and the adjacent myocardial wall. In contrast, laminin, an ECM protein highly expressed in the basement membrane of epithelia, was highly expressed in the ICVS at E10.5 but was down-regulated by E11.5-E12.5 (*Figure 4P–R*). Together, these data suggest that epithelial SHF progenitors in the ICVS differentiate to valve interstitial cells, but not via an undifferentiated mesenchymal intermediate. To rule out EMT as a mechanism, we evaluated

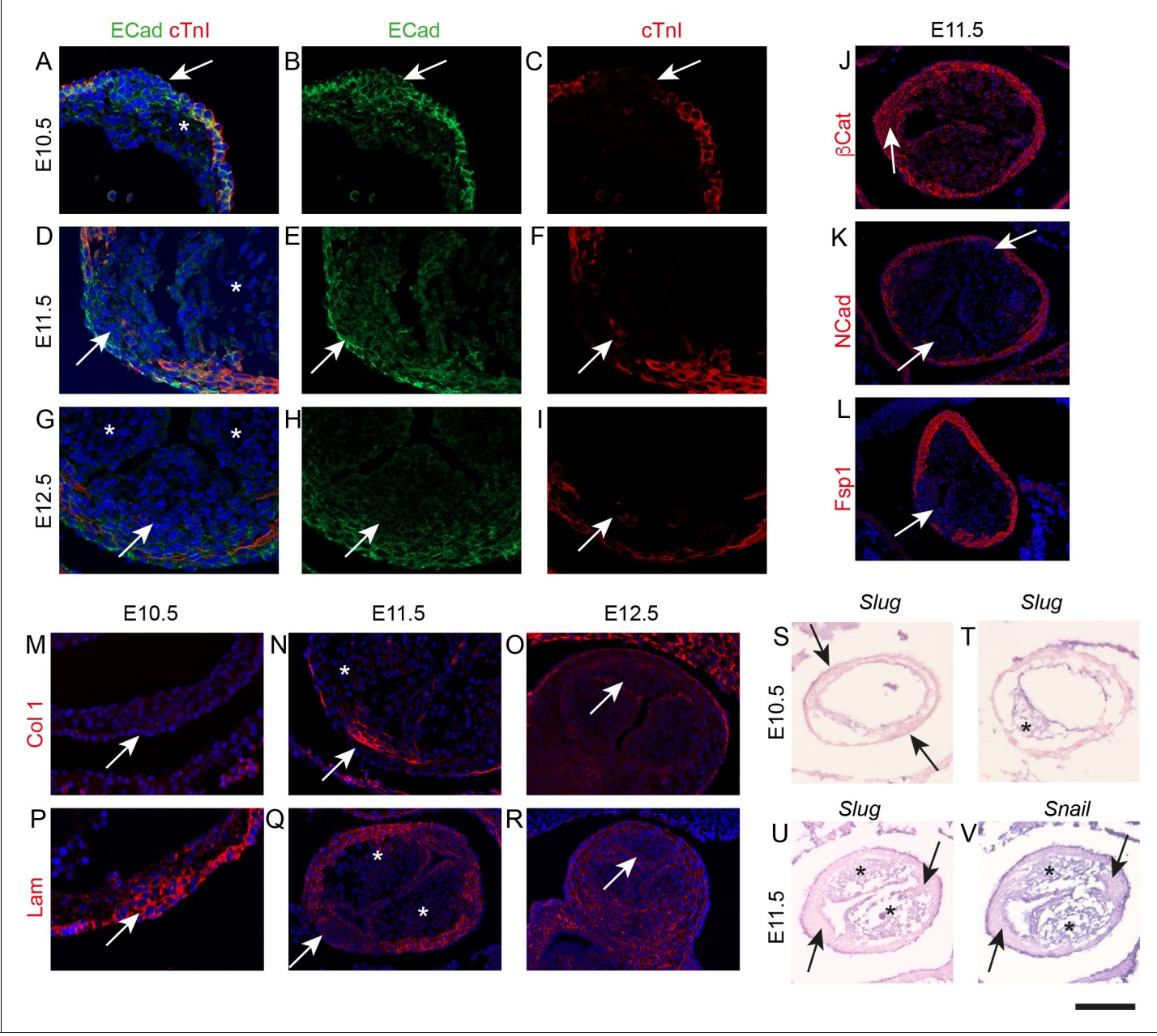

**Figure 4.** The ICVS transition from epithelium to mesenchyme, but not by a typical process of EMT. Unless otherwise stated, the aortic IC is shown. Arrows point to the ICVS or posterior valve primordium in each case. * mark the main outflow cushions. Images are typical examples from a minimum of n = 3 at each stage. (**A–C**) The ICVS express E-cadherin (green) at E10.5 (**A,B**), as does the surrounding myocardial outflow wall (labelled red by cTnI), although the main cushions do not. (**D–I**) E-cadherin is downregulated in the ICVS at E11.5 (**D,E**), and is no-longer expressed by E12.5 (**G,H**). (**J–L**) β-catenin is expressed in the ICVS and the surrounding wall at E11.5 (**J**), but neither N-cadherin nor Fsp1 (**K,L**) are found in the ICVS at this stage. (**M–O**) Collagen-I is transiently up-regulated specifically in the ICVS but not the main cushions at E11.5, but resembles the main leaflet primordia with little if any expression at E10.5 and E12.5. (**P–R**) Laminin is also transiently expressed in the ICVS at E10.5, but is downregulated and lost from the valve primordia by E12.5. (**S–V**) Neither *Slug* nor *Snail* are expressed in the ICVS (arrows in S,U,V) at E10.5–11.5, although they are both abundant in the main cushions (* in T,U,V). Scale bar: A-I,M,N,p=60 μm, J-L,O,Q-V = 100 μm.

DOI: https://doi.org/10.7554/eLife.34110.011

The following figure supplement is available for figure 4:

**Figure supplement 1.** ICVS do not express classical mesenchymal markers as they form.

DOI: https://doi.org/10.7554/eLife.34110.012

expression of *Slug* and *Snail*, key regulators of EMT, by in-situ hybridisation, focussing on E10.5 and E11.5 embryos, as by E11.5 Sox9 is found in the ICVS (*Figure 2Q,T*) and thus the valve cell precursors have formed. Notably, there was no expression of transcripts for either *Slug* or *Snail* within the ICVS at E10.5 or E11.5, although both were abundantly expressed in both the endothelium and newly formed mesenchyme of the main outflow cushions (*Figure 4S–V* and *Figure 4—figure supplement 1*). Thus, although the cells of the ICVS change from an epithelial to valve interstitial cell phenotype, this does not occur via classical EMT.

## Jagged-Notch signalling is active and required for ICVS development

Notch1 signalling plays an essential role in formation of the atrioventricular and main outflow cushions in the developing heart. It is involved, together with its ligand Dll4, in EndMT (*Luna-Zurita et al., 2010*), but also acts later during valve remodelling with an alternative ligand, Jag1 (*MacGrogan et al., 2016*). Moreover, *NOTCH1* mutations have been implicated in BAV (*Garg et al., 2005*). To identify the molecular mechanism underpinning formation of the ICVS we first asked whether there is Notch1 signalling activity in the ICVS. Immunolabelling confirmed that the active Notch1 receptor (Notch1 intracellular domain or N1ICD) was present in the non-myocardial walls of the aortic sac, the distal outflow and the endocardium. It was specifically expressed in cells within the ICVS at E10.5 although by E11.5 only a few cells in the cardiac jelly overlying the core of the ICVS continued to express nuclear N1ICD (*Figure 5A,E,I*), indicating a temporal window of Notch1 signalling activity. Immuno staining for Notch2 (using an antibody that recognised the uncleaved form of the protein) was also expressed in the membranes of cells within the ICVS but not in the main cushions (*Figure 5B,F,J*). Dll4 was found in the endocardium with N1ICD, but not in the ICVS at either E10.5 or E11.5 (*Figure 5—figure supplement 1*). We also evaluated the expression of both Jag1 and Jag2 ligands in this region of the heart. Jag1 protein was specifically detected within the distal outflow wall and in the core of the ICVS at both E10.5 and E11.5 (*Figure 5C,G,K*). In contrast, Jag2 was detected only in the inner adluminal region of the ICVS, close to the Notch1ICD-expressing endothelial cells (*Figure 5D,H,L*). On closer examination Notch 2, Jag1 and Jag2 were found at the membrane of ICVS cells that expressed Isl1 at E10.5 and E11.5 whereas N1ICD was mostly expressed in the nucleus of Isl1-negative cells (*Figure 5Q*-AB and *Figure 5—figure supplement 2*). By E12.5, neither N1ICD or Notch2, nor Jag1 or Jag2, were expressed in the interstitial cells within the ICVS (*Figure 5M–P*), although both N1ICD and Notch2 were expressed in the valve endothelium. Thus, it appeared that Notch signalling was temporally and spatially associated with differentiation of ICVS SHF-derived cells into valve interstitial cells.

To test the requirement for Jag-Notch signalling in formation of the ICVS, we deleted Jag1 gene expression using *Tnnt2-Cre* and examined the arterial valve appearances at E16.5. No *Jag1^f/f^;Tnnt2-Cre* (0/11) or *Jag2^f/f^;Tnnt2-Cre* (0/2) embryos demonstrated BAV, nor were abnormalities observed in *Tnnt2-Cre*-negative embryos (*Figure 6A*). We therefore wondered if Jag1 and Jag2 might be compensating for each other in the formation of the ICVS, as has been demonstrated in ventricular maturation (*D'Amato et al., 2016*). *Jag1^f/f^*, *Jag2^f/f^* and *Tnnt2-Cre* animals were inter-crossed and the arterial valve phenotype examined in the resulting offspring. In this complex genetic breeding experiment, BAV was observed in 1/3 *Jag1^f/+^;Jag2^f/f^;Tnnt2-Cre* mutants examined (*Figure 6B*), whilst in the absence of both ligands 4/11 *Jag1^f/f^;Jag2^f/f^cTnnt2-Cre* embryos demonstrated BAV (*Figure 6C*). Notably, the posterior (non-coronary) leaflet appeared to be missing in each case and there was no obvious presence of a raphe that would indicate leaflet fusion. One additional *Jag1^f/f^;Jag2^f/f^;Tnnt2-Cre* embryo had only a rudimentary posterior leaflet (*Figure 6D*) and a further 4/11 *Jag1^f/f^;Jag2^f/f^;Tnnt2-Cre* embryos had thickened or dysplastic arterial valve leaflets affecting either/both the aorta and pulmonary valve (*Figure 6C*). In these latter cases, abnormalities in all three leaflets of the valves were apparent. Thus, 9/11 *Jag1^f/f^;Jag2^f/f^;Tnnt2-Cre* embryos examined had major abnormalities in the development of the arterial valve leaflets, that affected the anterior/posterior leaflets that develop from the *Tnnt2-Cre*+ ICVS. Most (7/9) of these valve defects in the *Jag1^f/f^;Jag2^f/f^;Tnnt2-Cre* embryos occurred in the context of other cardiac anomalies, most commonly ventricular septal defect or double outlet right ventricle.

Similar abnormalities to those we observed in *Jag2^f/f^;Tnnt2-Cre* mutants have previously been described for the Notch regulator *Mib1^f/f^;Tnnt2-Cre* embryos (*Captur et al., 2016*). We therefore confirmed these findings, showing absence (1/9) or dysplasia (8/9) of the non-adjacent (non-coronary) leaflet in the mutant embryos at E16.5 (*Figure 6E,F*). We then examined the formation of the

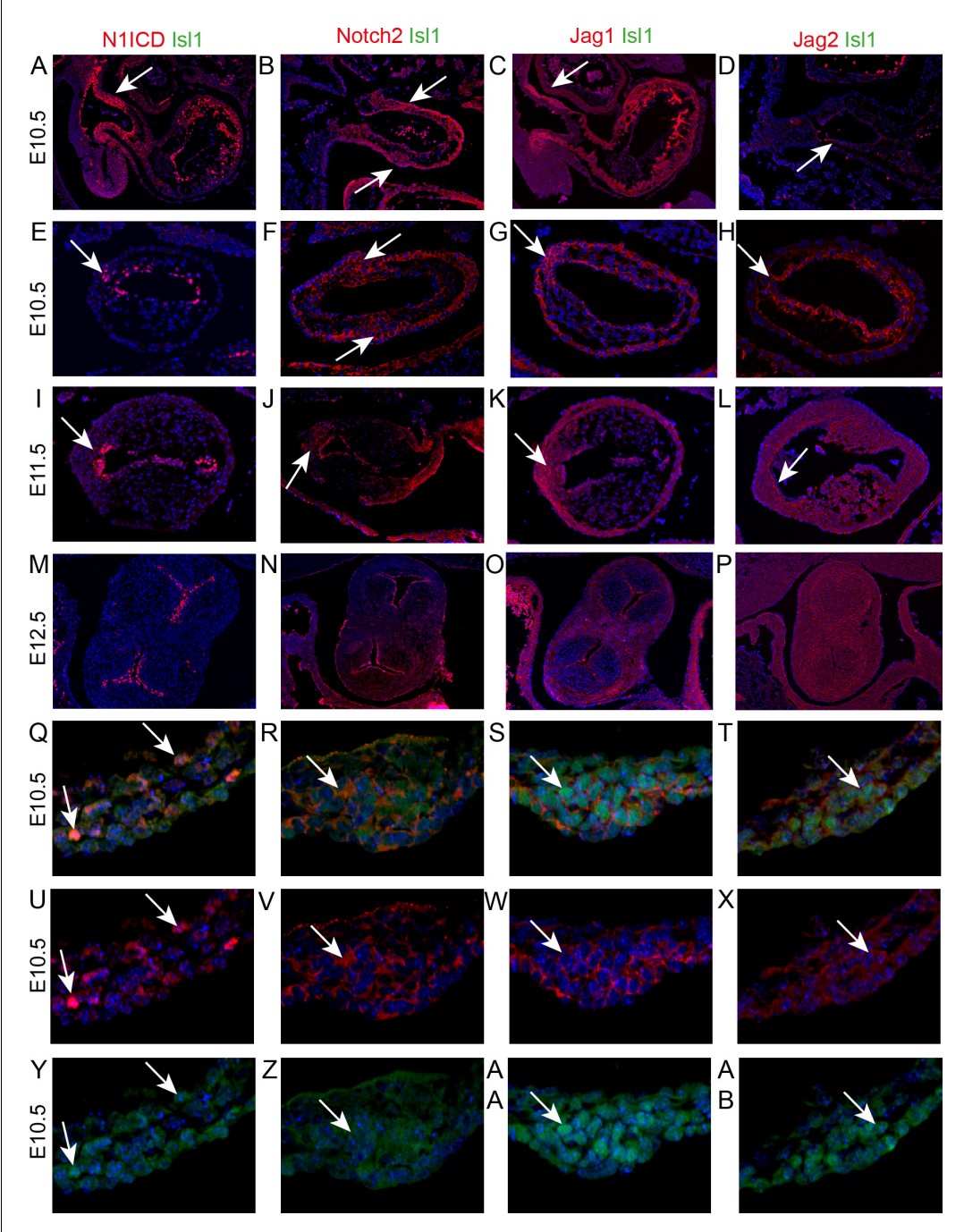

**Figure 5.** Notch signalling is found in the developing ICVS. All sections are frontal unless otherwise stated. Images are typical examples from a minimum of n = 3 at each stage. In E-AB, arrows point to the ICVS. (A–D) Sagittal sections show that Notch1ICD, Notch2 and Jag1 are abundant in the distal outflow wall (arrows) at E10.5. Jag2 is only found in the endothelium (arrow in D). (E–L) Notch1ICD, Notch2, Jag1 and Jag2 are found in the endocardium overlying the main cushions, but not within the cushions themselves at E10.5–11.5. All four localise to the ICVS at E10.5, with Notch1ICD, Notch2 and Jag1 maintained at E11.5. (M–P) Notch1ICD, Notch2 and Jag1 are found in the valve endothelium at E12.5, but none of these, nor Jag2, is found in the valve interstitial cells G-AB) Isl1 and Notch1ICD co-localise in the nuclei of only a few cells within the ICVS at E10.5, although Notch2, Jag1 and Jag2 are found in the membrane of Isl1 +nuclei containing cells in the core of the ICVS at this stage. Scale bar: A-D = 360 µm, E-H = 120 µm, I-L = 150 µm, M-P = 225 µm, Q-AB = 75 µm.

DOI: https://doi.org/10.7554/eLife.34110.013

The following figure supplements are available for figure 5:

**Figure supplement 1.** Dll4 immuno-reactivity is found in the endocardium from E10.5-E12.5 (arrows).

*Figure 5 continued on next page*

Figure 5 continued

DOI: https://doi.org/10.7554/eLife.34110.014

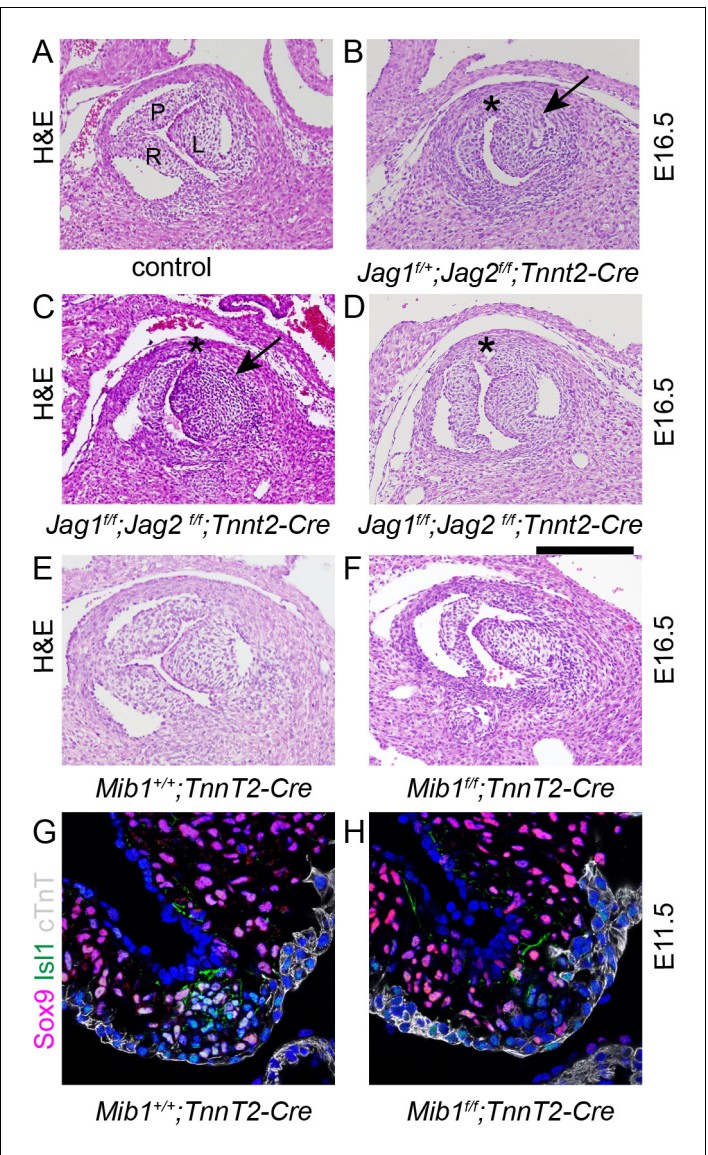

**Figure 6.** Disruption of Notch signalling results in abnormalities of the posterior (non-coronary) leaflet. Transverse H and E-stained sections of E16.5 embryos from *Jag1^f;Jag2^f;Tnnt2-Cre* litters. Images are typical examples. (**A–D**) Whereas the control embryo has three thinned leaflets in its aortic valve (**A**), the *Jag1^{f/+;}Jag2^{f/f};Tnnt2-Cre* and *Jag1^{f/f};Jag2^{f/f};Tnnt2-Cre* embryo (**B,C**) have only two thickened leaflets with no leaflet in the posterior position (*). In the *Jag1^{f/f};Jag2^{f/f};Tnnt2-Cre* embryo (**D**) a rudimentary posterior leaflet can be seen. The arrows point to the thickened left leaflet. (**E,F**) The *Mib1^{f/f};Tnnt2-Cre* embryo also has a hypoplastic posterior leaflet and dysplasia of the other valve leaflets. (**G,H**) Examination of the forming ICVS at E11.5 shows a reduction in the number of Sox9+/Isl1 +cells in the *Mib1^{f/f};Tnnt2-Cre* embryo compared to its control littermate. Images are typical examples from a minimum of n = 3 at each stage. A - anterior, L – left, R – right. Scale bar: A-F = 200 μm, G,H = 60 μm.
DOI: https://doi.org/10.7554/eLife.34110.016

ICVS in these mutants at E11.5, showing that in 1/3 of the *Mib1^{f/f};Tnnt2-Cre* examined, the ICVS were poorly formed with fewer Sox9+/Isl1 +cells (*Figure 6G,H*). This supports the idea that Notch signalling plays a critical role in the formation of the ICVS.

## SHF deficiency leads to ICVS defects and BAV

These data indicate that the SHF addition to the outflow tract wall is required for formation of the ICVS, which then give rise to the posterior leaflet of the aortic valve and the anterior leaflet of the pulmonary valve. We wondered if reduction of SHF addition to the ICVS would be sufficient to create hypoplastic ICVS and produce BAV. We have previously disrupted SHF addition to the outflow tract by deleting *Vangl2* gene expression from the SHF using *Isl1-Cre*. This resulted in disruption of epithelial polarity and premature differentiation of SHF cells within the distal outflow tract of *Vangl2; Isl1-Cre* mutants, leading to double outlet right ventricle and ventricular septal defect (*Ramsbottom et al., 2014*). At that time, we did not examine the development of the aortic and pulmonary valves. As the naturally occurring *Vangl2* mutant *loop-tail*, which is fully re-capitulated by SHF deletion of *Vangl2* (*Ramsbottom et al., 2014*), demonstrates aortic, but not pulmonary, hypoplasia (*Henderson et al., 2001*), we speculated that loss of *Vangl2* in the SHF progenitors would disrupt formation of the ICVS, particularly affecting the aortic side of the outflow. Analysis of seven *Vangl2^{f/f}-Isl1-Cre* embryos at E15.5 revealed that 57% (4/7) of them had dysplastic leaflets and BAV without raphe, with absence of the leaflet normally positioned posteriorly (*Figure 7A–D* and *Figure 7—figure supplement 1*). Dysplasia of the pulmonary valve was also observed in these hearts. Analysis at E10.5 and 11.5 showed that although both the aortic and the pulmonary ICVS were present in *Vangl2^{f/f};Isl1-Cre* heart, they were misplaced. In 2/6 cases, they were also smaller than expected (*Figure 7E–H*). Despite this they demonstrated apparently normal expression of a panel of markers at E11.5, including Isl1, β-catenin and E-cadherin (*Figure 7E–H* and *Figure 7—figure supplements 1* and *2*). However, Notch1ICD and Jag1 were missing or reduced in 2/3 E11.5 *Vangl2^{f/f}; Isl1-Cre* hearts examined (*Figure 7I–L*), suggesting that Notch signalling was disrupted in these mutants. Moreover, there was abnormal distribution of cTnI-expressing myocardial cells surrounding the ICVS at E11.5 and E12.5 (arrows in *Figure 7—figure supplements 1* and *2*). Close examination suggested that these were within the ICVS/valve primordium and so likely reflect abnormal differentiation or ingrowth of cardiomyocytes rather than detachment of layers of myocardium. Together, the hypoplastic ICVS and misexpression of cTnI suggest that rather than forming valve interstitial cells, the ICVS may be mis-specified or prematurely differentiate to myocardium, thus preventing the formation of a mature leaflet in this position.

DiGeorge syndrome is a chromosomal micro-deletion syndrome with cardiovascular defects, due to deficiency of the *TBX1* gene. *Tbx1* null mice replicate the cardiovascular features seen in patients with DiGeorge syndrome. As the SHF-derived pulmonary myocardium of the outflow tract is deficient in these mice, and absence of the anterior leaflet of the pulmonary valve has also been reported (*Théveniau-Ruissy et al., 2008*), we wondered whether these mutants might exhibit hypoplasia of the pulmonary ICVS. Analysis of *Tbx1* null mice at E15.5 and E12.5 confirmed that the valve leaflets were markedly abnormal within the unseptated *Tbx1^{-/-}* outflow tract (*Figure 8A,D*). Utilising αSMA and cTnI antibodies, a single ICVS could be seen in the *Tbx1^{-/-}* outflow tract at E12.5. Looking earlier, at E11.5, whereas a well-developed aortic ICVS could be seen in *Tbx1^{-/-}*, it was impossible to distinguish a discrete pulmonary ICVS, with cTnI and αSMA-expressing cells within the expected region (*Figure 8E–H*). Thus, as in *Vangl2;Isl1-Cre* mutants, the disrupted development of the ICVS may underpin, at least in part, the arterial valve defects seen in *Tbx1* mutant mice.

## Discussion

We have shown that the formation of the arterial valves, whilst encompassing some processes that are similar to the atrioventricular valves, is uniquely different, by virtue of the large influx of NCC and the dominant role played by the SHF. The central four valve leaflets, formed as the outflow tract septates, are filled by NCC and SHF cells secondarily derived through endocardium and transformed into valve interstitial cell precursors by EndMT. In contrast, the intercalated valve structures are not cardiac jelly-derived endocardial cushions, but instead differentiate directly from (*Mef2c-AHF-Cre+; Tnnt2-Cre+*) SHF-derived outflow wall; *Figure 9*). These cells retain signatures of previous cardiac troponin promoter activation, although they are not, and never have been, of myocardial phenotype.

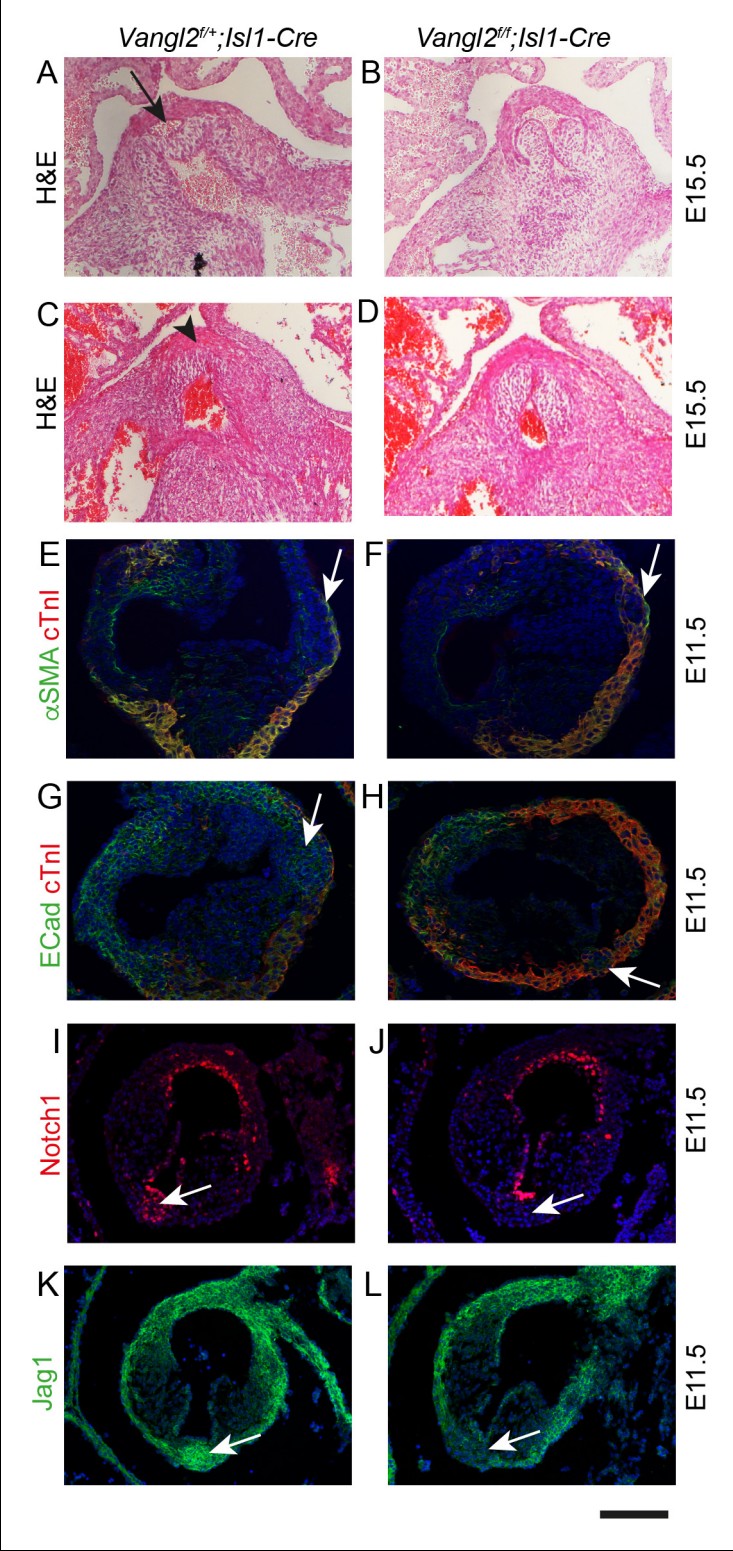

**Figure 7.** Failure to form the aortic posterior (non-coronary) leaflet in *Vangl2^{f/f};Isl1Cre* embryos. Images are typical examples of a minimum of n = 3. (**A–D**) Whereas the posterior leaflet (arrow in A) can clearly be seen in the control embryo at E15.5, this is missing in the *Vangl2^{f/f};Isl1-Cre* embryos (arrowhead in C). (**E–H**) Hypoplastic and malpositioned ICVS (arrows) can be seen in *Vangl2^{f/f};Isl1-Cre* embryos at E11.5. Expression of outflow markers appears otherwise normal. (**I–L**) Notch1ICD and Jag1 expression levels are reduced in the ICVS of some *Vangl2^{f/f};*
*Figure 7 continued on next page*

*Figure 7 continued*

*Isl1-Cre* embryos compared to control littermates at E11.5 (arrows), although there is normal expression elsewhere in the outflow walls. Scale bar: A-D = 120 μm, E-H = 100 μm, I-L = 120 μm.

DOI: https://doi.org/10.7554/eLife.34110.017

The following figure supplements are available for figure 7:

**Figure supplement 1.** ICVS formation in *Vangl2;Isl1*-Cre mutants.

DOI: https://doi.org/10.7554/eLife.34110.018

**Figure supplement 2.** Marker expression in *Vangl2;Isl1*-Cre mutant ICVS.

DOI: https://doi.org/10.7554/eLife.34110.019

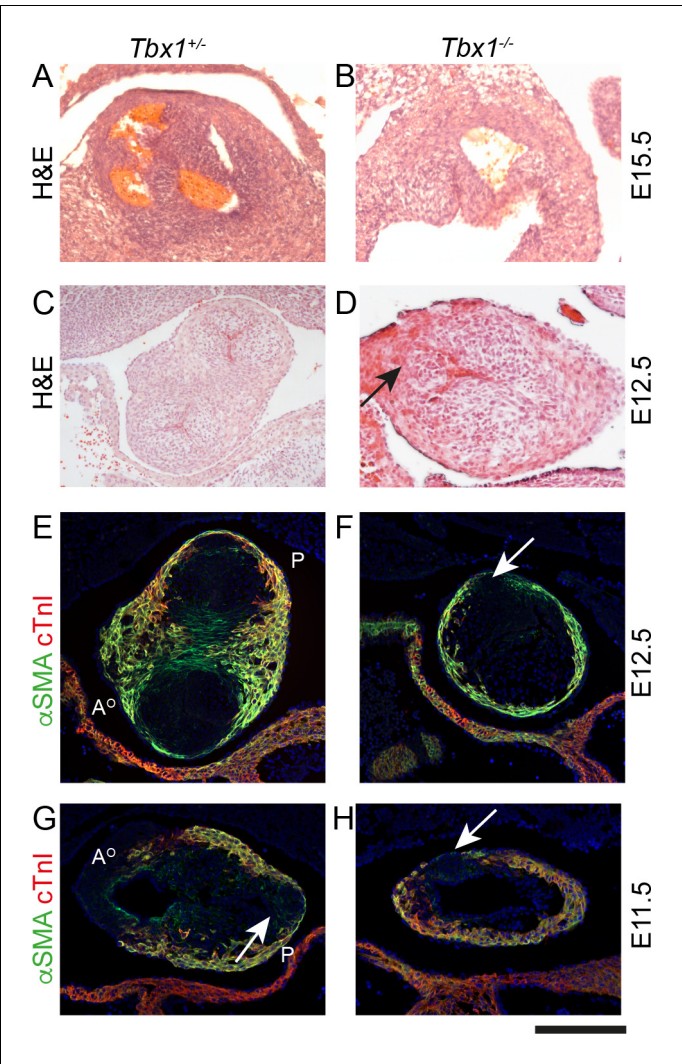

**Figure 8.** Failure to form the pulmonary anterior leaflet in *Tbx1* null embryos. Images are typical examples of a minimum of n = 3. (**A,B**) The outflow tract remains unseptated and the valve leaflets are markedly abnormal in the *Tbx1*⁻/⁻ at E15.5. It is not possible at this stage to establish the identity of the leaflets. (**C–F**) At E12.5, *Tbx1*⁺/⁻ embryos have septated their outflow tract and have three leaflet primordia in both the aorta and pulmonary trunk. In contrast, the *Tbx1*⁻/⁻ littermate has not septated its outflow tract and there is a single ICVS (arrows in D and F). (**G,H**) At E11.5, the aortic and pulmonary ICVS are obvious and are distinct from the myocardial wall (labelled yellow by αSMA/cTnI dual staining) in the *Tbx1*⁺/⁻ embryo. In contrast, only a single ICVS (arrow) is seen in the *Tbx1*⁻/⁻ embryo. Scale bar: A = 150 μm, B-F = 100 μm.

DOI: https://doi.org/10.7554/eLife.34110.020

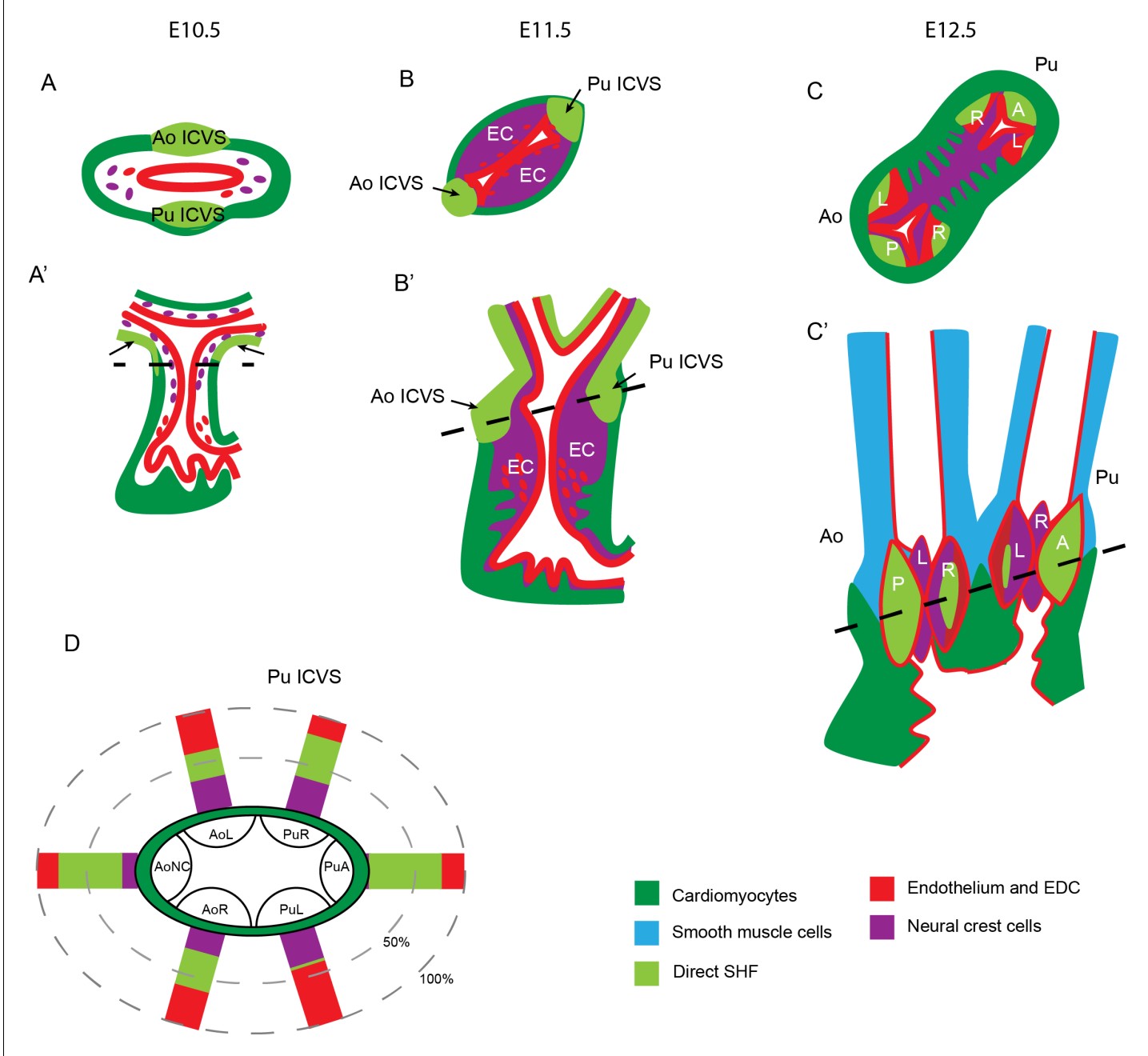

**Figure 9.** Summary of lineage contributions to the aortic and pulmonary valve leaflets. (**A–C**) Cartoon showing the development of the ICVS and A/P valve leaflets in frontal (**A–C**) and transverse (**A'–C'**) orientation from E10.5-E12.5, superimposed with the contributions from each lineage of cells. Arrows point to the ICVS except in A' where these cannot be seen (as they are found in the inferior and superior walls) and the arrows point to the distal additions of SHF cells to the lateral walls. Dotted lines in A'-C' illustrate the plane of section shown in A-C. The ICVS appear as swellings within the outer wall of the distal outflow tract at E10.5, expand by E11.5 and discrete from the wall by E12.5, resembling the other valve leaflet primordia. (**G**) Cartoon summary of the contributions of NCC, EDC and the population derived directly from the SHF to the valve leaflet primordia in the normal embryo at E12.5. A – anterior leaflet, Ao – aorta, AoNC – aortic non-coronary (posterior) leaflet, EC – endocardial cushion, ICVS – intercalated valve swelling, L- left leaflet, Pu – pulmonary, R – right leaflet.

DOI: https://doi.org/10.7554/eLife.34110.021

As the ICVS mature, they switch on valve leaflet markers such as Sox9 and Versican, without utilising EndMT as a mechanism or expressing typical mesenchymal markers. Ultimately, however, they appear indistinguishable from the valve leaflets derived from the main outflow cushions. Specifically, we find no evidence of trans-differentiation of either cardiomyocytes or SMC to the interstitial cells within the ICVS, the concomitant reduction of Isl1 and increase in Sox9 protein levels within the ICVS instead indicating a mechanism of direct differentiation of SHF progenitors into valve cells.

These studies show that three lineages make significant contributions to the arterial valve leaflets; NCC, EDC and a novel population of SHF progenitors cells that transiently activate the *Tnnt2-Cre* driver, but differentiate directly into valve interstitial cells without passing through the endocardium nor myocardial lineages. This novel population makes the major contribution to the anterior and posterior leaflets derived from the ICVS, and also significant contributions to the left and right leaflets, derived from the main cushions (*Figure 9*). Although a technical limitation of these Cre-based studies is the inability to track both SHF and NCC lineages concomitantly, the absolute numbers of cells counted, and the proportions of each, suggest that any other lineages can only comprise a minor component (*Figure 9*). Previous studies have suggested that the possibility of trans-differentiation of cardiomyocytes in the distal outflow (*Ya et al., 1998*). These studies were performed before knowledge of the SHF and the now accepted concept that cells are added to the distal poles of the outflow tract. Our data strongly argue against trans-differentiation as a mechanism for formation of the ICVS as although the *Tnnt2-Cre+* cells activated the cTnT promoter, we never saw co-expression of cardiomyocyte markers in the forming ICVS and could identify a few *Tnnt2-Cre+/cTnI-* cells even at E9.5, before the ICVS form. As well as making the major contribution to the forming ICVS, these *Tnnt2-Cre+* lineage cells contribute cells to the postnatal valve leaflets although detailed phenotyping and analysis of the fate of these cells is beyond the scope of this manuscript.

Cre lineage tracing identifies cells and their descendants that currently, or have ever, expressed the promoter/enhancer element used to drive Cre protein-based recombination of a reporter gene. It is evident from this study that to interpret such data it is essential to track protein expression of the *Cre*-driven promoter product. In the original description of the *Tnnt2-Cre* driver (*Jiao et al., 2003*), it was stated that the rat cTnT promoter construct drove cardiomyocyte-specific expression up to E10.5, but its expression pattern later in gestation was not described. It has also been reported that cTnT is expressed in some SMC and can modulate their contraction (*Wang et al., 2001*; *Kajioka et al., 2012*). From data presented here it is clear that *Tnnt2-Cre* labels arterial SMC, as well as the ICVS. However, as we could not find evidence of cTnT protein expression in these latter cell types, even when we looked at the earliest stages, it is likely that transient activation of the promoter occurred in undifferentiated, potentially multipotent, SHF progenitors within the distal outflow. Analysis of this phenomenon in relation to the SMC is the subject of further study in our laboratory. With respect to the ICVS, rather than becoming cardiomyocytes, *Tnnt2-Cre* labeled cells activated Sox9 expression and differentiated to valve interstitial cells, although this occurred at E11.5, approximately a day later than the more proximal cells within the wall differentiated to cardiomyocytes.

Our data are generally in agreement with and build on the data of Sizarov et al (*Sizarov et al., 2012*). However, these authors did not observe Sox9 staining within the cells of the ICVS of approximately stage-matched human embryos, although they did see Isl1 labelling. They also suggested that the 'columns' they described were mesenchymal, although our data show that, at least in the early stages, they are epithelial. Moreover, we did not see expression of classical mesenchymal markers such as N-cadherin, αSMA and Fsp1, although the ICVS did express elastin, which is found in mature valve leaflets. The differences between these studies are unclear, but it may be that they reflect the transient expression of some of these markers and/or slight differences in staging and maturation of the ICVS between mouse and human. The delay in differentiation of cells within the ICVS may at least partially explain the activation of the *Tnnt2* promoter in these cells, as they may be poised, but stalled, for differentiation. The presence of *Tnnt2-Cre+* lineage cells in the ICVS of the mouse may be a valuable tool to investigate other genes involved in ICVS development, as it allows deletion in the directly-differentiating SHF population in the ICVS, without affecting NCC or EDC. For example, Mib1 is a ubiquitin ligase required in signalling cells for Jag activity. Here, and in a previous publication (*Captur et al., 2016*), arterial valve defects were observed in *Mib1;Tnnt2-Cre* mutants, although initially the reason for this was unclear. Our data suggest that these defects likely arise because *Tnnt2-Cre* is activated in the valve primordia, deleting *Mib1* and thus inactivating

Jag1/Jag2-Notch signalling. This prevents the ICVS, and to a lesser extent the main cushions, from forming properly thus resulting in abnormal leaflets in the mutant embryos.

EndMT is a crucial process for typical endocardial cushions, but not for ICVS, where the major process is development from SHF cells that directly differentiate into valve interstitial cells in situ. In contrast, there is minor deposition of ECM and Sox9+ cells in the sub-endothelial compartment of the forming ICVS, indicative of a small contribution by EndMT to the ICVS in this region. Notch1 together with Dll4, plays a critical early role during EndMT and Notch signalling is known to be vital for normal development of the main endocardial cushions. Deletion of both *Notch1* and *Dll4* (*Limbourg et al., 2005*), result in the early death of the embryos, by E10.5, precluding examination for ICVS formation and thus making it difficult to exclude EndMT from a crucial role in formation of the ICVS. However, evidence from the knockout of *Brg1* in endothelium using *Nfatc1-Cre* (*Akerberg et al., 2015*) provides evidence that ICVS are not dependent on EndMT as both anterior and posterior arterial valves leaflets appeared well formed in this mutant.

Our mouse data, in keeping with those of Sizarov who studied human embryos (*Sizarov et al., 2012*), suggest that the ICVS form as expansions of compacted cells within the wall of the distal OFT from cells in continuity with the epithelial SHF. These condensations form at the proximal extent of the 'fishmouth' of non-myocardial distal outflow wall (*Anderson et al., 2012*) as seen in *Figure 1O*, likely reflecting the asymmetrical addition of SHF cells and the rotation of the outflow wall (*Scherptong et al., 2012*; *Bajolle et al., 2006*). These cell condensations then separate from the surrounding wall, forming discrete structures that then undergo differentiation to typical valve cells. Thus, these ICVS cells appear to share an origin and be juxtaposed with the myocardial component of the outflow wall. Indeed, both the myocardial and the ICVS cells are *Tnnt2-Cre+* suggesting a close relationship between the two populations. It is of note that Jag1 was abundant in the ICVS, which do not utilise EndMT/EMT, and is also abundantly expressed in the outflow tract myocardium. Although Jag1 is also essential for valve development, its role in EndMT is controversial, with some authors suggesting it is important (*Wang et al., 2013*), but others suggesting it is not required for EndMT but plays a later role in the remodelling phases of the main cushion-derived heart valves (*MacGrogan et al., 2016*). From our data, and those of others (*High et al., 2007*), it seems likely that it is Notch1 and/or Notch2 that is the main Notch receptor in the ICVS. Notch1ICD was found in the distal outflow wall and the ICVS, but only transiently, and it was largely down-regulated by E11.5. Notch2 was expressed throughout the ICVS and until E11.5, although as we could not obtain an antibody for the active form of this protein, we cannot be sure that signalling was underway throughout this period. This transient expression within the ICVS is in contrast to the endocardium overlying the main cushions, where both Notch1 and Notch2 were maintained until E12.5 and beyond (*Del Monte et al., 2007*). Notch signalling has been shown to be important for the differentiation of cardiac progenitor cells, with its down-regulation crucial for differentiation to cardiomyocytes (*Nemir et al., 2006*; *Kwon et al., 2009*) and for activation of Sox9 in cartilage precursors (*Mead and Yutzey, 2009*). Thus, the presence and then subsequent removal of Notch1 signalling may be essential in the ICVS for their differentiation to valve interstitial cells. In support of the idea that Notch-Jag signalling is playing an important role in the development of the ICVS, the deletion of *Jag1/Jag2* using *Tnnt2-Cre* resulted in abnormal development of the arterial valve leaflets in 9/11 double mutants examined, including BAV without raphe. Specifically, in 5/11 cases the phenotype was absence or hypoplasia of the non-coronary leaflet that is derived from the ICVS. Although the *Jag1/2;Tnnt2-Cre* mice have a range of defects affecting the ventricular myocardium (*D'Amato et al., 2016*), and so valve defects secondary to the ventricular defects cannot strictly be ruled out, it seems more likely that the reduction/loss of the non-coronary leaflet results from the deletion of Jag expression specifically in the ICVS. The redundancy of *Jag1* and *Jag2* is intriguing and is reminiscent of ventricular wall development (*D'Amato et al., 2016*). Although Jag1 was expressed in the core of the ICVS, Jag2 was restricted to the adluminal cells, close to N1ICD-expressing cells, suggesting that there may be some ligand-specific functional specialization. Further studies will be necessary to delineate these likely complex interactions.

The *Vangl2;Isl1-Cre* and *Tbx1* mutant mouse lines, both of which have abnormalities associated with limited addition of SHF cells to the outflow, have abnormal valve leaflets that result from failure to form or mature the ICVS; a mechanism that has not previously been described. In the *Vangl2* mutant, premature differentiation of SHF to cardiomyocytes, as occurs earlier in the aortic sac (*Ramsbottom et al., 2014*), may reduce the availability of SHF-derived cells to form Sox9+ valve

interstitial cells and may underlie the reduction in the size of the ICVS and ultimately lead to BAV. The reduction in Notch1ICD and Jag1 in 2 of 3 *Vangl2^{f/f};Isl1-Cre* embryos suggests that Notch signalling may be playing a role in this process and be regulated by *Vangl2*, either directly or indirectly. Notably, the defects were more common in the aortic valve (as they are in the Notch signalling pathway mutants) and although the reasons for this are uncertain, aortic (but never pulmonary) hypoplasia is common in the naturally occurring *Vangl2* mutant, *loop-tail* (*Henderson et al., 2001*). This suggests that it is the SHF that contributes to the aorta that is preferentially affected in this mutant. Premature loss of progenitor phenotype and differentiation of cardiomyocytes leading to a marked reduction of SHF-derived subpulmonary myocardium has been reported in the *Tbx1* mutant mouse (*Vitelli et al., 2010*). In addition, absence of the anterior leaflet of the pulmonary valve (derived from the pulmonary ICVS) has also been reported (*Théveniau-Ruissy et al., 2008*). Our analyses extend these data to show that a discrete pulmonary ICVS is never formed in the *Tbx1^{-/-}* and highlights similarities in the ICVS phenotype between these and the *Vangl2;Isl1-Cre* mutants. Moreover, the role of both *Vangl2* and *Tbx1* in regulating the epithelial properties of the SHF (*Ramsbottom et al., 2014*; *Francou et al., 2014*), which form the myocardium and also appear to form the ICVS, suggests that the epithelial phenotype of these cells, at least at early stages of their development, may be crucial. Indeed, it raises the possibility that there is an intrinsic relationship between the loss of epithelial properties of the wall and differentiation of the cells within it to cardiomyocytes. These differences between the affected valve observed between *Vangl2* and *Tbx1* mutants likely reflect inherent differences in the patterning of the outflow tract, further reflecting subdomains within the SHF (*Vitelli et al., 2010*; *Scherptong et al., 2012*). BAV has been described in a number of mouse mutants. Notably, in several cases (including following knockout of *Alk2*, a BMP receptor (*Thomas et al., 2012*), and Slit/Robo (*Mommersteeg et al., 2015*) signalling) the anterior/non-coronary leaflets specifically failed to form properly. This, therefore may be a common mechanism underlying BAV (without raphe) formation, at least in mice, and may occur when there is disruption of the SHF as we see in the *Vangl2;Isl1-Cre* and *Tbx1* mutants. Raphe are infrequently described in mouse mutants with BAV, although this may be because most mouse mutants are described during fetal life when the leaflets remain unsculpted and thus before a true raphe would be apparent. Although it is not possible to demonstrate the mechanisms of arterial valve development using human embryos it is important to note that at comparable stages of development the appearances of human and mouse valve primordia are very similar and importantly, the expression pattern of Isl1, marking SHF progenitors, is identical. Together our data suggest that loss of SHF progenitors, either because of a failure of their formation or their premature differentiation, may be a common cause of BAV and can explain the common association between this and other intra-cardiac heart malformations. As such, BAV formed in this way may link to the subtype of BAV patterns without raphe, as occurs in 5–10% of BAV cases (*Sievers and Schmidtke, 2007*; *Buchner et al., 2010*). It will be important to analyse genomic data from patients with this BAV phenotype, searching in particular in SHF and Jag-Notch pathways.

Our murine and human data show differences between the structure of the aortic and pulmonary ICVS as they develop, that could have relevance to the common incidence of BAV compared to bicuspid pulmonary valve. The aortic ICVS develops more proximally and is not surrounded by a myocardial cuff as it develops, compared with the pulmonary ICVS, which is more distal and is completely surrounded by cardiomyocytes. These differences have also been reported recently in a histological examination of human embryos (*Milos et al., 2017*). Moreover, the pulmonary ICVS has a lesser contribution from NCC than does the aortic, and based on the differences we see between *Vangl2* and *Tbx1* mutants, may have different genetic regulation. The ICVS also express tropoelastin during their formation, whereas the main cushions do not, suggesting that these differences in cell lineage between the forming leaflets are reflected by expression of specific proteins. These differences are likely to have morphogenetic impacts on the developing aortic and pulmonary ICVS, which might reflect in their differential predisposition to pathology. Although bicuspid pulmonary valve is rare as an isolated finding in humans, it is common in association with tetralogy of Fallot (*Chacko et al., 2017*). Thus, defects in the SHF that result in failure to develop one or other ICVS might be relevant to a high proportion of human valve abnormalities.

# Materials and methods

**Key resources table**

| Reagent type (species) or resource | Designation | Source or reference | Identifiers | Additional information |
|---|---|---|---|---|
| Genetic reagent (Mus musculus) | Mef2c-AHF-Cre | PMID: 16188249 | | |
| Genetic reagent (Mus musculus) | Wnt1-Cre | PMID: 9843687 | | |
| Genetic reagent (Mus musculus) | Tek-Cre | PMID: 11161575 | | |
| Genetic reagent (Mus musculus) | Wt1-ERT-Cre | PMID: 18568026 | | |
| Genetic reagent (Mus musculus) | Tnnt2-Cre | PMID: 12975322 | | |
| Genetic reagent (Mus musculus) | ROSA-Stop-eYFP | PMID: 11299042 | | |
| Genetic reagent (Mus musculus) | Isl1-Cre | PMID: 14667410 | | |
| Genetic reagent (Mus musculus) | Vangl2 flox | PMID: 25521757 | | |
| Genetic reagent (Mus musculus) | mTmG | PMID: 17868096 | | |
| Genetic reagent (Mus musculus) | Tbx1 null | PMID: 11242110 | | |
| Genetic reagent (Mus musculus) | Jag1 flox | PMID: 15550486 | | |
| Genetic reagent (Mus musculus) | Jag2 flox | PMID: 20533406 | | |
| Genetic reagent (Mus musculus) | Mib1 flox | PMID: 18043734 | | |
| Biological sample(Homo sapien) | Embryo sections CS14, 16, 21 | HDBR; PMID: 26395135 | | |
| Antibody | Anti-GFP | Abcam | ab13970 | 1/150 |
| Antibody | Sox9 | Abcam | ab185230 | 1/150 |
| Antibody | Versican | Abcam | ab1032 | 1/150 |
| Antibody | Collagen I | Abcam | ab21286 | 1/150 |
| Antibody | Alpha smooth muscle actin | Abcam | ab5228 and ab5694 | 1/150 |
| Antibody | SM22 alpha | Abcam | ab14106 | 1/150 |
| Antibody | Cardiac troponin T | Abcam | ab8295 | 1/150 |
| Antibody | E-cadherin | Abcam | ab15148 | 1/150 |
| Antibody | Notch2 | Abcam | ab8926 | 1/150 |
| Antibody | Cre recombinase | NEB | 15036 | 1/150 |
| Antibody | Cardiac troponin I | HyTest | 4T21/2 | 1/150 |
| Antibody | Laminin | Sigma | L9393 | 1/150 |
| Antibody | CS56 | Sigma | C8035 | 1/150 |
| Antibody | Isl1 | Developmental Studies Hybridoma Bank | 39.4D5 and 40.2D6 | 1/150 |
| Antibody | N-cadherin | BD Transduction Laboratories | 610920 | 1/150 |
| Antibody | Notch1ICD | Cell Signalling | 4147 | 1/150 |
| Antibody | Jagged 1 | Cell Signalling | 2620 | 1/150 |

*Continued on next page*

*Continued*

| Reagent type (species) or resource | Designation | Source or reference | Identifiers | Additional information |
|---|---|---|---|---|
| Antibody | Jagged 2 | BioRad | MCA5708GA | 1/150 |
| Sequence -based reagent | Snail cDNA probe for in situ hybridisation | PMID: 9671584 | | |
| Sequence -based reagent | Slug cDNA probe for in situ hybridisation | this paper | | sequences provided in text |

## Mouse strains and histological analysis

CD1 mice were used to observe normal histology of the developing heart. Mef2c-*AHF-Cre* (*Verzi et al., 2005*), *Wnt1-Cre* (*Danielian et al., 1998*), *Tek-Cre* (*Kisanuki et al., 2001*), *Wt1-ERT-Cre* (*Zhou et al., 2008*) and *Tnnt2-Cre* (*Jiao et al., 2003*), intercrossed with the *ROSA-Stop-eYFP* (*Srinivas et al., 2001*) or *mTmG* (*Muzumdar et al., 2007*), were used to follow cells of the required lineage/cell type. As both marker genes are targeted into the Rosa26 locus and there is no suggestion of differences in recombination in response to Cre between these two lines (*Liu et al., 2013*), the two lines were used interchangeably. *Vangl2$^{flox}$;Isl1-Cre* (*Ramsbottom et al., 2014*), *Tbx1* (*Jerome and Papaioannou, 2001*), *Jag1$^{flox}$;Jag2$^{flox}$;Tnnt2-Cre* (*D'Amato et al., 2016*), and *Mib1$^{flox}$; Tnnt2-Cre* (*Garg et al., 2005*) were used and genotyped as described previously. Timed matings were carried out overnight and the presence of a copulation plug was designated embryonic day (E) 0.5. All mice were maintained on a C57Bl/6 background, backcrossed for three generations, then maintained by brother-sister matings. Littermate controls were used where appropriate. Each experiment was carried out in, at minimal, triplicate. Cre expression was induced in the Wt1-ERT-Cre line by Tamoxifen injection. Pregnant females were injected intraperitoneally with 100 µl of tamoxifen (10 mg/ml) dissolved in ethanol and peanut oil on three consecutive days (E9.5, E10.5, E11.5). Embryos were then collected at E12.5. Mice were maintained according to the Animals (Scientific Procedures) Act 1986, United Kingdom, under project license PPL 30/3876. All experiments were approved by the Newcastle University Ethical Review Panel.

Embryos were harvested at different developmental stages, rinsed in ice-cold phosphate-buffered saline (PBS) and fixed in 4% paraformaldehyde at 4°C, the duration of which depended on the age of the embryo (4 hr for E9.5–10.5, overnight for E11.5 and 48 hr for E12.5 to E15.5), before paraffin embedding. Embryos were embedded in either a transverse, frontal or sagittal orientation. For each analysis carried out using embryos, a minimum of 3 embryos for each developmental stage and section orientation were used (biological replicates). Stage matching was carried out based on somite counting and other external features such as limb maturity. The analyses were carried out as separate experiments and a minimum of 5–10 sections (technical replicates) were analysed for consistency. For basic histological analysis, paraffin-embedded embryos were sectioned and stained with haemotoxylin and eosin following standard protocols.

## Human embryos

The human embryonic and fetal material was provided by the Joint MRC/Wellcome Trust (grant # 099175/Z/12/Z) Human Developmental Biology Resource (www.hdbr.org [*Gerrelli et al., 2015*]). For basic histological analysis, paraffin-embedded embryos were sectioned and stained with haemotoxylin and eosin following standard protocols.

## Three-dimensional reconstruction

Serial sections were photographed from control and mutant embryos at E10.5 and E11.5. The sections were aligned and three dimensional reconstructions were carried out using Amira 4.0 according to standard protocols (*Soufan et al., 2003*).

## Immunohistochemistry/Immunofluorescence

Methodology has been published previously (*Phillips et al., 2007*). Sections were cut from paraffin embedded embryos at 8 µm using a rotary microtome (Leica). Slides were de-waxed with Histoclear (National Diagnostics) and rehydrated through a series of ethanol washes. For immunohistochemistry for lineage tracing purposes, endogenous peroxidases were inhibited by treating slides with 3%

H$_2$O$_2$ for 5 min. Following washes in PBS, antigen retrieval was performed by boiling slides in citrate buffer (0.01 mol/L) pH 6.3 for 5 min. Samples were blocked in 10% FCS and then incubated overnight at 4°C with the primary antibody GFP (Abcam) diluted in 2% FCS. Samples were then incubated with an anti-chicken biotinylated secondary antibody (Dako) for 1 hr, then with AB complex (Vector labs) for 30 min then stained with DAB and mounted using Histomount. This amplification step ensured maximal detection of antigen. Bright field images were captured using the Zeiss Axioplan. For immunofluorescence, following washes in PBS, antigen retrieval was performed by boiling slides in citrate buffer (0.01 mol/L) pH 6.3 for 5 min. Sections were blocked in 10% FCS and then incubated overnight at 4°C with the primary antibodies (see key resources table) diluted in 2% FCS. After washing, sections were incubated at room temperature for one hour, with secondary antibodies conjugated to either Alexa 488 or Alexa 594 (Life Technologies). Fluorescent slides were then mounted with Vectashield Mounting medium with DAPI (Vector Labs). Each experiment was carried out in triplicate using separate embryos (>3) and imaging multiple sections of the structure of interest (>5). Immunofluorescence images were collected with using a Zeiss Axioimager Z1 fluorescence microscope equipped with Zeiss Apotome 2 (Zeiss, Germany). Acquired images were processed with AxioVision Rel 4.9 software.

### cDNA probes and in situ hybridisation

Methodology was adapted from (Smart et al., 2002). Digoxigenin (DIG)-labeled RNA antisense probes corresponding to the protein coding regions of *Slug* (EF450456.1; Forward primer: TTTCAACGCCTCCAAGAAGC; Reverse primer: TAATACGACTCACTATAGGGGCAGCCAGACTCC TCATGTT) and *Snail* (NM_001032543; [Sefton et al., 1998]) were synthesized by T7 RNA polymerase (MAXIScript, Ambion) and in vitro transcription (pGEM_T easy, Promega), respectively. The slide in situ hybridization was performed using 10 um paraffin-embedded sections of E10.5 and E11.5 mouse embryos. Slides were de-waxed with Histoclear (National Diagnostics) and rehydrated with ethanol series washes. Slides were then fixed in 4% DEPC-PFA for 20 min at room temperature prior to the treatment with proteinase K (10 µg/mL) for 10 min at 37°C. Post-fixation by 4% DEPC-PFA was performed at room temperature for 5 min. Slides were then processed to the acetylation reaction for 10 min in solution containing 100 mM Triethanolamine, 40 mM Hydrochloric acid and 250 mM Acetic Anhydride. After that, pre-hybridization was performed for 2 hr at 68°C using hybridization buffer (50% formamide, 5x SSC (pH6), 200 µg/mL T-RNA, 100 µg/mL Heparin, 0.1% tween-20); hybridization was carried out for 16 hr using the same buffer with RNA probes. Following a series of washes, slides were blocked in 2% MABT (maleic acid buffer containing Tween 20) blocking solution (Roche) for 2 hr and incubated with anti-digoxigenin-AP, fab fragments (Roche) overnight at 4°C. Slides were then developed in NBT/BCIP solution for 18–36 hr at room temperature and the reaction terminated through incubation in 100% methanol for 5 min. Counterstaining with 1% eosin for 5 min was performed, followed by dehydration with washes in a serial ethanol dilutions (30–60 s per wash), incubation in Histoclear twice for 10 min, and finally mounting slides with Histomount (Agar scientific). Each experiment was carried out in triplicate using separate embryos (>3) and imaging multiple sections of the structure of interest (>5).

### Cell counts and statistical analysis

For quantification of *Mef2c-AHF-Cre* in the E12.5 arterial valve leaflet primordia, labelled and unlabelled cells were counted by a single observer (with checks for accuracy from an independent observer) from each of the six discrete primordia. In each case, three representative sections from three different embryos were counted. All sections were 8 µm (as were comparable data sets in Phillips et al., 2013). Chi squared analysis was used to show that the proportion of *Mef2c-AHF-Cre* lineage cells between the different arterial valve primordia were significantly different. Pairwise chi-squared comparison between the primordia, with Bonferroni correction for multiple testing, confirmed that the aortic non-coronary primordium was statistically different from all other primordia.

## Acknowledgements

The British Heart Foundation (PG/15/46/31589 and RG/12/15/29935), together with SAF2016.78370-R, CB16/11/00399 and RD16/0011/0021 from the Spanish Ministry of Science, Industry and Universities, funded this study.

# Additional information

## Funding

| Funder | Grant reference number | Author |
|---|---|---|
| British Heart Foundation | RG/12/15/29935 | Lorriane Eley<br>Rachel V Richardson<br>Lindsay Murphy<br>Bill Chaudhry<br>Deborah J Henderson |
| British Heart Foundation | PG/15/46/31589 | Lorriane Eley<br>Bill Chaudhry<br>Deborah J Henderson |
| Ministerio de Ciencia, Innovación y Universidades of Spain | CB16/11/00399 (Ciber Cardiovascular) | Donal MacGrogan<br>Alejandro Salguero-Jimenez<br>José Luis de La Pompa |
| Ministerio de Ciencia, Innovación y Universidades of Spain | SAF2016-78370-R | Donal MacGrogan<br>Alejandro Salguero-Jimenez<br>José Luis de La Pompa |
| Ministerio de Ciencia, Innovación y Universidades of Spain | RD16/0011/0021 (Red de Terapia Celular, TERCEL) | Donal MacGrogan<br>Alejandro Salguero-Jimenez<br>José Luis de La Pompa |

The funders had no role in study design, data collection and interpretation, or the decision to submit the work for publication.

## Author contributions

Lorriane Eley, Data curation, Formal analysis, Investigation, Writing—review and editing; Ahlam MS Alqahtani, Rachel V Richardson, Lindsay Murphy, Alejandro Salguero-Jimenez, Marcos Sintes Rodriguez San Pedro, Shindi Tiurma, Lauren McCutcheon, Adam Gilmore, Investigation; Donal MacGrogan, Formal analysis, Investigation, Writing—review and editing; José Luis de La Pompa, Resources, Supervision, Funding acquisition, Writing—review and editing; Bill Chaudhry, Conceptualization, Resources, Data curation, Formal analysis, Supervision, Funding acquisition, Investigation, Visualization, Methodology, Writing—original draft, Writing—review and editing; Deborah J Henderson, Conceptualization, Resources, Data curation, Formal analysis, Supervision, Funding acquisition, Investigation, Visualization, Methodology, Writing—original draft, Project administration, Writing—review and editing

## Author ORCIDs

José Luis de La Pompa  https://orcid.org/0000-0001-6761-7265
Bill Chaudhry  http://orcid.org/0000-0003-2833-8882
Deborah J Henderson  http://orcid.org/0000-0002-2705-5998

## Ethics

Human subjects: Human embryos were obtained from the MRC/Wellcome funded Human Developmental Biology Resource (HDBR) held at Newcastle University, UK. Full ethical approval was obtained by this resource.
Animal experimentation: Embryos were then collected at E12.5. Mice were maintained according to the Animals (Scientific Procedures) Act 1986, United Kingdom, under project license PPL 30/3876. All experiments were approved by the Newcastle University Ethical Review Panel.

## Decision letter and Author response

Decision letter https://doi.org/10.7554/eLife.34110.024
Author response https://doi.org/10.7554/eLife.34110.025

## Additional files

**Supplementary files**
• Transparent reporting form
DOI: https://doi.org/10.7554/eLife.34110.022

**Data availability**
All data are included in the manuscript and supplementary files.

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
