## [Decision Letter]

Thank you for submitting your article "Direct differentiation of second heart field progenitors to arterial valve cells linked to bicuspid aortic valve" for consideration by *eLife*. Your article has been favorably evaluated by Didier Stainier (Senior Editor) and three reviewers, one of whom is a member of our Board of Reviewing Editors. The reviewers have opted to remain anonymous.

The reviewers have discussed the reviews with one another and the Reviewing Editor has drafted this decision to help you prepare a revised submission.

Summary:

Eley et al. present evidence that intercalated outflow tract cushions form from second heart field derived mesenchyme. In a clearly written manuscript the authors suggest that these cells arise from undifferentiated cells in the outflow wall through Notch signaling. Defining the cellular origins of valve mesenchyme, defects in which cause valve dysmorphogenesis and disease, is an important objective. Differences in mesenchymal cell composition between the aortic and pulmonary ICs and right and left aortic and pulmonary valve primordia are documented, identifying a previously poorly characterized arterial valve population that appears to originate in the SHF. While this detailed study potentially furthers our understanding of arterial valve formation, the reviewers consider that the following revisions are required in order to justify the authors' conclusions.

Essential revisions:

1) While the authors have carried out extensive genetic tracing experiments, the lineage data claiming the SHF origin of arterial valve cells is not compelling. Additional Cre based genetic cell tracing experiments with better spatiotemporal specificity are required to draw this conclusion. The major limitation is the non-inducible *Tnnt2-Cre* approach that would ideally be resolved by using inducible *Tnnt2-Cre* lines (Wu et al. Genesis 2009 or Yan et al. Genesis 2015). Unless this data is available, the authors should temper their conclusion, including in the Abstract and title. The authors should also evaluate Cre protein expression to define the precise expression profile of *Tnnt2-Cre* in the intercalated valve primordia.

2) The authors should investigate whether Isl1 and Sox9 are transiently co-expressed in cells in the ICs at E11.5 and E12.5. This would provide a strong supporting argument and distinguish their observations from the prior results of Sisarov and colleagues (2012) who showed that human IC primordia express ISL1 and have low levels of SOX9 expression, in contrast to the major cushions (see Figure 5 in Sizarov et al.). The authors also need to discuss the findings of Sisarov et al. in the light of the new data.

3) The authors argue that IC interstitial cells form from the SHF epithelium, although not through EMT, based on analysis of epithelial marker distribution and lack of EMT transcription factor expression. The manuscript would be significantly strengthened if they could demonstrate that EMT is not required for formation of IC mesenchyme. For example, the authors should show that IC formation is normal in mouse embryos in which the process of EndMT is perturbed (such as endocardial Notch conditional mutant embryos). In addition, please better define an alternate mechanism of mesenchymal cell production that is hinted at. For example, Figure 4 and the cartoon in Figure 9 suggest that this may occur by a local thickening of the outflow wall (see also Sisarov et al.).

4) Co-immunostaining for NICD (as well as membrane NOTCH1-4) and Jag1 or Jag2 or Isl1 in low and high magnifications is recommended to provide a better view of NOTCH activation in the forming ICVS valves. It would be helpful to perform *Tnnt2*+ lineage analysis of ICVS valves in Jag1/2 compound mutant mice to define the cellular defect(s) underlying the underdeveloped non-coronary valve leaflet.

5) Given the common failure of posterior IC leaflet development in Notch and *Vangl2* conditional mutant hearts, the authors should provide information as to the epistatic relationship between Notch and *Vangl2*. In the *Vangl2* mutant the OFT myocardial wall appears to detach in the innermost layers in Figure 7—figure supplement 2; this defect in wall integrity should also be commented on.

Points 6-8 require clarifications/modifications of the text rather than new experiments.

6) Please clarify that the results pertain to cases of BAV without a raphe in the Introduction, Discussion and title. In particular, the authors should discuss whether, as their results imply, BAV without a raphe is more frequently observed linked with outflow CHD.

7) Concerning developmental asymmetry between the aortic and pulmonary IC, please qualify in the manuscript, including the Abstract, whether activated Notch expression and the functional requirement for Notch in IC formation is specific to the posterior (non-coronary) aortic IC (as suggested in Figures 5 and 6). The authors should also discuss why IC formation may be restricted to particular circumferential regions of the OFT wall, including discussion of prior evidence for differences between the aortic and pulmonary sides of the OFT (Bajolle et al., 2008; Scherptong et al., 2012).

8) Concerning the last section of the Results, which is less developed, Eley et al. should state that *Tbx1*, like *Vangl2*, is required to regulate normal epithelial features of SHF cells as well as evidence from Théveniau-Ruissy et al., 2008 (Figure 3 therein) that the pulmonary IC is hypoplastic at early developmental stages. New insights provided by the data in Figure 8 beyond those from Théveniau-Ruissy et al., 2008 should be clearly indicated.

---

## [Author Response]

Essential revisions:1) While the authors have carried out extensive genetic tracing experiments, the lineage data claiming the SHF origin of arterial valve cells is not compelling. Additional Cre based genetic cell tracing experiments with better spatiotemporal specificity are required to draw this conclusion. The major limitation is the non-inducible Tnnt2-Cre approach that would ideally be resolved by using inducible Tnnt2 Cre lines (Wu et al. Genesis 2009 or Yan et al. Genesis 2015). Unless this data is available, the authors should temper their conclusion, including in the Abstract and title. The authors should also evaluate Cre protein expression to define the precise expression profile of Tnnt2-Cre in the intercalated valve primordia.

*Mef2c-AHF-Cre* is highly specific to SHF-derived tissues within the heart. Similarly, Isl1 antibody reliably labels cells within the outflow tract that are derived from the SHF and still retain progenitor characteristics. For this reason, we believe that there is good evidence that the cells that are found within the ICVS are derived from the SHF and remain undifferentiated for longer that the surrounding tissues (also derived from the SHF). However, in line with the reviewer’s request, we have changed the title and been more circumspect in the Abstract.

We understand the reviewers’ comments about the timing of the labelling of the cells by *Tnnt2-Cre*. Unfortunately we do not have the inducible *Tnnt2-Cre* line available to us, and it would not have been possible to obtain the line and optimise its use within the proposed period for the resubmission, however, as suggested, we been able to pin down further the window when the *TnnT2-Cre* was active. *Tnnt2-Cre* has been shown to be expressed in the cardiac crescent from E7.5 and is retained in cardiomyocytes throughout embryonic life (Jiao et al., 2003). These authors also showed that it was effective in deleting floxed allelles after E9.5, but that earlier than this, deletion was incomplete. This is likely because during the period of E8.5 and E10.5, SHF cells are adding to the OFT and differentiating (thus initiating *Tnnt2* expression) only once they are within it. By E10.5 the *TnnT-Cre* driven *YFP* expression is found more distally than is cTnT protein within the outflow wall (Figure 3T-V). This suggests that the *TnnT2-Cre* may be recombining in differentiating cells that do not yet express differentiated cardiomyocyte markers. E10.5 is the earliest time point when we can identify the *TnnT2-Cre*-labelled intercalated valve swellings within the outflow tract and at this stage they have already downregulated cardiomyocyte markers (Figures 2 and 3). As suggested by the reviewers, we have now used an anti-Cre antibody to pin down the period when Cre protein is present. This has shown that whilst Cre protein is present in the ICVS at E10.5, it is not found at E11.5 and E12.5. We have been unable to find any data about the perdurance of the Cre protein, and anyway this is likely to be cell type and temporally specific. However, taken together, this data is consistent with labelling of the ICVS prior to E10.5 (as no cTnT protein is found in the ICVS at this stage), at the time of, or just prior to, their progenitors in the SHF entering the OFT. This data is now included in new Figure 3—figure supplement 2 and is described in the first paragraph of the subsection “Origin of ICVS directly from SHF progenitors in the outflow wall”.

2) The authors should investigate whether Isl1 and Sox9 are transiently co-expressed in cells in the ICs at E11.5 and E12.5. This would provide a strong supporting argument and distinguish their observations from the prior results of Sisarov and colleagues (2012) who showed that human IC primordia express ISL1 and have low levels of SOX9 expression, in contrast to the major cushions (see Figure 5 in Sizarov et al.). The authors also need to discuss the findings of Sisarov et al. in the light of the new data.

We agree with the reviewers that this is a critical point. We have thus carried out co-localisation studies with Isl1 and Sox9 at E11.5 and E12.5 and confirmed that there is transient co-expression of the two markers in ICVS cells. This data is included in Figure 3W-Y and Figure 3—figure supplement 4 and is discussed in the last paragraph of the subsection “Origin of ICVS directly from SHF progenitors in the outflow wall”. We have also discussed the relevance of this data and how it relates to the previous study by Sizarov et al. (2012) in the fourth paragraph of the Discussion.

3) The authors argue that IC interstitial cells form from the SHF epithelium, although not through EMT, based on analysis of epithelial marker distribution and lack of EMT transcription factor expression. The manuscript would be significantly strengthened if they could demonstrate that EMT is not required for formation of IC mesenchyme. For example, the authors should show that IC formation is normal in mouse embryos in which the process of EndMT is perturbed (such as endocardial Notch conditional mutant embryos). In addition, please better define an alternate mechanism of mesenchymal cell production that is hinted at. For example, Figure 4 and the cartoon in Figure 9 suggest that this may occur by a local thickening of the outflow wall (see also Sisarov et al.).

We agree with the reviewers that this is a critical point. In the first instance we have clarified that there is likely a minor early contribution of EndMT to the developing ICVS, as some of the cells closest to the lumen appear to be endocardial in origin (*Tek-Cre* positive – Figure 3D) and we also observed deposition of ECM molecules and Sox9+ cells next to the endocardium of the forming ICVS at E10.5, but not in the rest of the ICVS (arrowheads in Figure 2D, G, J, M, P, S). This appears to be a component of the cardiac jelly that lines the whole of the outflow tract and thus also overlies the forming ICVS. This is clarified in the subsection “IC undergo late differentiation to become valve primordia” and in the fifth paragraph of the Discussion.

The question of how we could address this experimentally is more complex. Notch1 KO mice die at approximately E10.5 (Limbourg et al., 2005), as do other mutants where EMT is prevented. Previously we attempted to create *Notch1^f/f^;Cad5-ERT-Cre* mice, however, optimisation of the Tamoxifen timing to create the desired phenotypes did not prove possible: when we treated with Tamoxifen from E8.5 the mice recapitulated the Notch1;*Tek-Cre* phenotype and died before E10.5; they did not have a well enough developed heart to see the ICVS. When Tamoxifen was given from E9.5, there was no reproducible valve phenotype. Although this may give circumstantial support for the idea that Notch1 is not required for formation of the ICVS, as they form after E9.5, this is inconclusive. We have searched the literature and have been unable to find any cases where EndMT is comprehensively disrupted and the embryos survive to be analysed at stages beyond E11.5. However, we did find a published paper that showed that when *Brg1* (a component of the BAF chromatin remodelling complex) is knocked out specifically in the endocardium using *Nfatc1-Cre*, EndMT was reduced and valve abnormalities (BAV) were observed (Akerberg et al., 2015). However, the non-coronary leaflet that derives from the ICVS was well formed in all cases shown, suggesting that this is not affected by the reduction in EndMT. This now discussed in the fifth paragraph of the Discussion.

We agree with the suggestion that we need to better explain our proposed mechanism for formation of the ICVS. This is now discussed in the sixth paragraph of the Discussion.

4) Co-immunostaining for NICD (as well as membrane NOTCH1-4) and Jag1 or Jag2 or Isl1 in low and high magnifications is recommended to provide a better view of NOTCH activation in the forming ICVS valves. It would be helpful to perform Tnnt2+ lineage analysis of ICVS valves in Jag1/2 compound mutant mice to define the cellular defect(s) underlying the underdeveloped non-coronary valve leaflet.

We agree that this is an important point. We have therefore carried out additional staining for Notch1ICD, Notch2, Jag1, Jag2 and Isl1 in the developing ICVS at E10.5. These are included in Figure 5 and in Figure 5—figure supplement 2. From the literature (High et al., 2008) it does not appear that Notch3 and 4 are expressed in the distal outflow walls and therefore we did not analyse these further.

To address the second point, that it would be helpful to perform *Tnnt2*+ lineage analysis of ICVS valves in Jag1/2 compound mutant mice to define the cellular defect(s) underlying the underdeveloped non-coronary valve leaflet, we have made some efforts to address this. Although we could not carry out these experiments in Jag1/Jag2 double mutants as this complex cross was not available, we had mentioned in the discussion that *Mib1^f/f^;Tnnt2-Cre* mutants also have abnormalities of the arterial valves, as published in Captur et al., 2015. We have therefore included images of the *Mib1^f/f^;Tnnt2-Cre* valves in Figure 6, but also included analysis of the forming ICVS in mutant and control ICVS at E11.5. This shows that the ICVS is poorly formed in the f/f embryos and that there are fewer Sox9+/Isl1+ cells present, supporting the idea that there is a primary abnormality in the formation of the ICVS in the *Mib1^f/f^;Tnnt2-Cre* embryos. This is described in the last paragraph of the subsection “Jagged-Notch signalling is active and required for ICVS development”.

5) Given the common failure of posterior IC leaflet development in Notch and Vangl2 conditional mutant hearts, the authors should provide information as to the epistatic relationship between Notch and Vangl2. In the Vangl2 mutant the OFT myocardial wall appears to detach in the innermost layers in Figure 7—figure supplement 2; this defect in wall integrity should also be commented on.

In order to address this issue, we have carried out additional staining of Notch1 and Jag1 in *Vangl2* mutants at E11.5. This has demonstrated that Notch signalling is disrupted in 2/3 of the *Vangl2^f/f^;Isl1-Cre* embryos examined, suggesting that *Vangl2* may be upstream of Notch in the distal outflow wall. This is described in the first paragraph of the subsection “SHF deficiency leads to ICVS defects and BAV”. Whether this is direct or not remains unclear.

Relating to detachment of the OFT myocardium, we do not interpret this as detachment, but rather interpret this as myocardial differentiation within the ICVS. This is clarified in the aforementioned paragraph.

Points 6-8 require clarifications/modifications of the text rather than new experiments.6) Please clarify that the results pertain to cases of BAV without a raphe in the Introduction, Discussion and title. In particular, the authors should discuss whether, as their results imply, BAV without a raphe is more frequently observed linked with outflow CHD.

We have clarified throughout that our data pertains to BAV without raphe. This is discussed in the seventh paragraph of the Discussion.

7) Concerning developmental asymmetry between the aortic and pulmonary IC, please qualify in the manuscript, including the Abstract, whether activated Notch expression and the functional requirement for Notch in IC formation is specific to the posterior (non-coronary) aortic IC (as suggested in Figures 5 and 6). The authors should also discuss why IC formation may be restricted to particular circumferential regions of the OFT wall, including discussion of prior evidence for differences between the aortic and pulmonary sides of the OFT (Bajolle et al., 2008; Scherptong et al., 2012).

We have added this information in the subsection “Jagged-Notch signalling is active and required for ICVS development”. We have also discussed inherent differences between the aortic and pulmonary sides of the outflow in the sixth and seventh paragraphs of the Discussion.

8) Concerning the last section of the Results, which is less developed, Eley et al. should state that Tbx1, like Vangl2, is required to regulate normal epithelial features of SHF cells as well as evidence from Théveniau-Ruissy et al., 2008 (Figure 3 therein) that the pulmonary IC is hypoplastic at early developmental stages. New insights provided by the data in Figure 8 beyond those from Théveniau-Ruissy et al., 2008 should be clearly indicated.

The role of *Tbx1* in regulating the epithelial properties of the SHF is discussed briefly in the seventh paragraph of the Discussion. We included additional text on the description of the abnormalities in the pulmonary ICVS in *Tbx1*-/- in the last paragraph of the subsection “SHF deficiency leads to ICVS defects and BAV”. We have also clarified that our study extends and explains the previous study by the Kelly group, by showing that the pulmonary ICVS never forms (general developmental delay in the *Tbx1*-/- means that there are no ICVS – including the aortic – at E10.5). However, the main purpose of this analysis in *Tbx1* mutants was to highlight the similarities to the *Vangl2* mutant and to highlight the importance of the epithelial SHF. This is now discussed in the seventh paragraph of the Discussion.